**Article** https://doi.org/10.1038/s41467-023-43584-6

# Higher Antarctic ice sheet accumulation and surface melt rates revealed at 2 km resolution

Brice Noël [1,2] ✉, J. Melchior van Wessem [2], Bert Wouters [3], Luke Trusel [4], Stef Lhermitte [3,5] & Michiel R. van den Broeke [2]

Antarctic ice sheet (AIS) mass loss is predominantly driven by increased solid ice discharge, but its variability is governed by surface processes. Snowfall fluctuations control the surface mass balance (SMB) of the grounded AIS, while meltwater ponding can trigger ice shelf collapse potentially accelerating discharge. Surface processes are essential to quantify AIS mass change, but remain poorly represented in climate models typically running at 25-100 km resolution. Here we present SMB and surface melt products statistically downscaled to 2 km resolution for the contemporary climate (1979-2021) and low, moderate and high-end warming scenarios until 2100. We show that statistical downscaling modestly enhances contemporary SMB (3%), which is sufficient to reconcile modelled and satellite mass change. Furthermore, melt strongly increases (46%), notably near the grounding line, in better agreement with in-situ and satellite records. The melt increase persists by 2100 in all warming scenarios, revealing higher surface melt rates than previously estimated.

The Antarctic ice sheet (AIS) is currently losing mass[1–3], as solid ice discharge, including ice shelf basal melting[4] and iceberg calving[1], exceeds the mass gained from surface processes. The surface mass balance (SMB), i.e., the difference between snow accumulation and surface ablation from sublimation, drifting snow erosion and meltwater runoff[5], primarily affects the synoptic, seasonal, interannual and decadal variability of AIS mass change[6]. Over floating ice shelves, surface melt triggers surface lowering, firn pore space depletion through retention and refreezing, and subsequent ponding of excess meltwater, that is eventually discharged into the ocean via crevasse and lake drainage, or runoff rivers[7–12]. As for Greenland fjords, (future) increase in surface and subsurface runoff have the potential to enhance sub-shelf basal melting and subsequent destabilisation by thinning ice shelves from below[4]. These combined processes, i.e., surface meltwater ponding and basal melt, can trigger ice shelf

disintegration by hydrofracturing, a mechanism that reduces the buttressing effect on the grounded AIS to eventually accelerate solid ice discharge and sea-level rise[13]. This has previously occurred in the Antarctic Peninsula over Larsen B ice shelf (March 2002)[14]. As surface melt is projected to increase exponentially with future atmospheric warming[15], it is expected that Antarctic ice shelves become increasingly prone to collapse[16,17].

Regional climate models and earth system models, that typically run at spatial resolutions of 25–100 km, are key tools to reconstruct the contemporary SMB and surface melt in Antarctica and project their future evolution[5,17–21]. However, their relatively coarse grids cannot accurately resolve the complex topography of e.g., the Antarctic Peninsula, the Transantarctic Mountains or Queen Maud Land, where both snowfall accumulation and melt gradients peak in response to steep slopes[22]. Consequently, climate models generally underestimate

[1]Laboratoire de Climatologie et Topoclimatologie, University of Liège, Liège, Belgium. [2]Institute for Marine and Atmospheric Research Utrecht, Utrecht University, Utrecht, Netherlands. [3]Department of Geoscience and Remote Sensing, Delft University of Technology, Delft, Netherlands. [4]Department of Geography, Pennsylvania State University, University Park, PA, USA. [5]Department of Earth and Environmental Sciences, KU Leuven, Leuven, Belgium. ✉e-mail: bnoel@uliege.be

orographic precipitation and propagate snowfall too far inland onto the AIS interior plateau[5,21]. A recent multi-model comparison revealed large inter-model annual SMB differences of 266 Gt year⁻¹ (~10% of the Antarctic-wide total)[5], that are primarily controlled by the grid resolution used in regional climate models[23]. Model evaluation using in situ measurements from automatic weather stations (AWS) (1992–2018)[24] and remote sensing from the satellite radar scatterometer (QuikSCAT) (2000–2009)[25] likewise suggests a general melt underestimation in e.g., the regional climate model RACMO2.3p2[18].

Here we present daily SMB and surface melt products covering the grounded AIS and floating ice shelves at 2 km spatial resolution for the contemporary climate and three scenario projections until 2100. As a first step, present-day climate from the global climate reanalysis ERA5[26] (1979–2021) and three global climate projections from the Community Earth System Model (CESM2)[27] under a low (SSP1-2.6), moderate (SSP2-4.5), and high-end (SSP5-8.5) warming scenario (1950–2099) are used as lateral forcing for the Regional Atmospheric Climate Model (RACMO2.3p2)[17,18], which simulates the contemporary and future SMB (components) of Antarctica on a 27 km grid (see Methods). As a second step, statistical downscaling is applied to correct these SMB components, including surface melt, runoff, total precipitation, snowfall, sublimation and snow drift erosion, for elevation biases between the relatively coarse RACMO2.3p2 grid at 27 km and a high-resolution surface topography from the Reference Elevation Model of Antarctica at 2 km (REMA)[28]. Melt and runoff are further adjusted for local albedo biases, e.g., regions exposing blue ice or darker bare ice mixed with rocks, not included in RACMO2.3p2, based on a 2 km albedo map from the Moderate Resolution Imaging Spectroradiometer (MODIS) averaged for 2000–2021 (see "Methods"). The ability of statistical downscaling to refine the spatial distribution of SMB components was first demonstrated for the Greenland ice sheet, where the downscaled product realistically captured high mass loss rates over narrow ablation zones and outlet glaciers that are typically unresolved in RACMO2.3p2[29]. Likewise, statistical downscaling to (sub-)kilometre spatial resolution proved essential to accurately quantify contemporary (and projected) mass change of the Greenland ice sheet[30,31], its peripheral ice caps[32], glaciers of the Canadian Arctic[33], Svalbard[34], and Iceland[35], and their contribution to global sea-level rise. Using in situ and remote sensing data for model evaluation, we show that our downscaled product for Antarctica at 2 km improves upon the original RACMO2.3p2 data at 27 km, by resolving SMB and surface melt patterns in unprecedented spatial detail, notably in topographically rough regions including mountain ranges and the vicinity of the grounding line.

## Results

### Larger accumulation rates across mountain ranges

Figure 1a, b shows the resulting contemporary Antarctic-wide SMB at 2 km and 27 km spatial resolution averaged for the period 1979–2021. Spatial differences are generally small (<200 mm w.e. year⁻¹) (Fig. 1c), except for the Antarctic Peninsula where precipitation peaks (Fig. 1d, e). Here, local differences can reach up to 2 m w.e. year⁻¹ (Fig. 1f). Note that as the extent of the 2 km and 27 km ice masks differ at the margins (Supplementary Fig. 1c, d), large local positive (resp. negative) differences occur when the 2 km ice mask covers larger (resp. smaller) areas than the 27 km product (Fig. 1f). Statistical downscaling overall enhances accumulation across high-elevated mountain ranges that are not properly resolved at 27 km resolution (Fig. 1c and Supplementary Fig. 1). In general, the orographic effect on precipitation is enhanced at 2 km resolution, with accumulation being higher on the steep, windward marginal slopes of the grounded AIS, and reduced towards the gently sloping inland plateau (Fig. 1c). In particular, accumulation is enhanced above the crest of the Antarctic Peninsula, with reductions relative to the 27 km grid at lower levels on both sides of the Peninsula and over Larsen C ice shelf (Fig. 1f), which is in line with the dry foehn winds that prevail in this region[36].

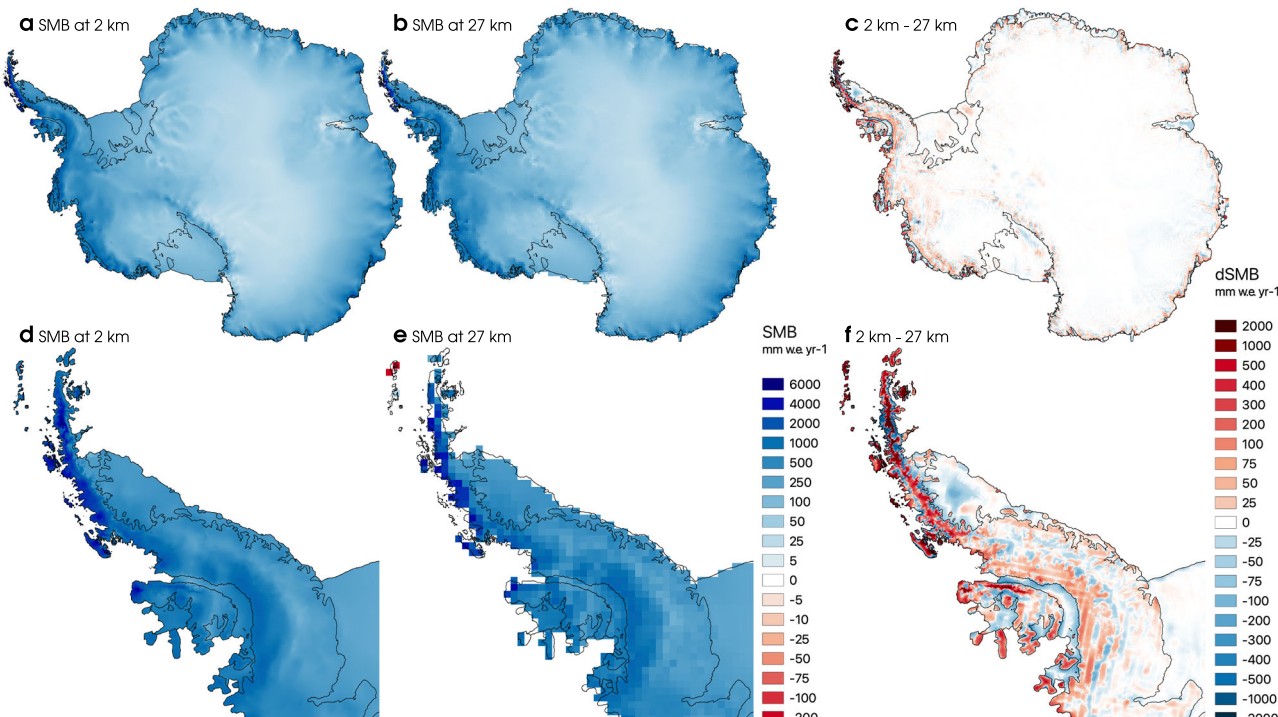

**Fig. 1 | High-resolution Antarctic surface mass balance.** Modelled mean annual Antarctic surface mass balance (SMB) for the period 1979–2021 (**a**) statistically downscaled to 2 km resolution, (**b**) modelled by RACMO2.3p2 at native 27 km resolution. (**c**) SMB difference between the 2 km and 27 km products. (**d**–**f**) Same as (**a**–**c**) but for the Antarctic Peninsula.

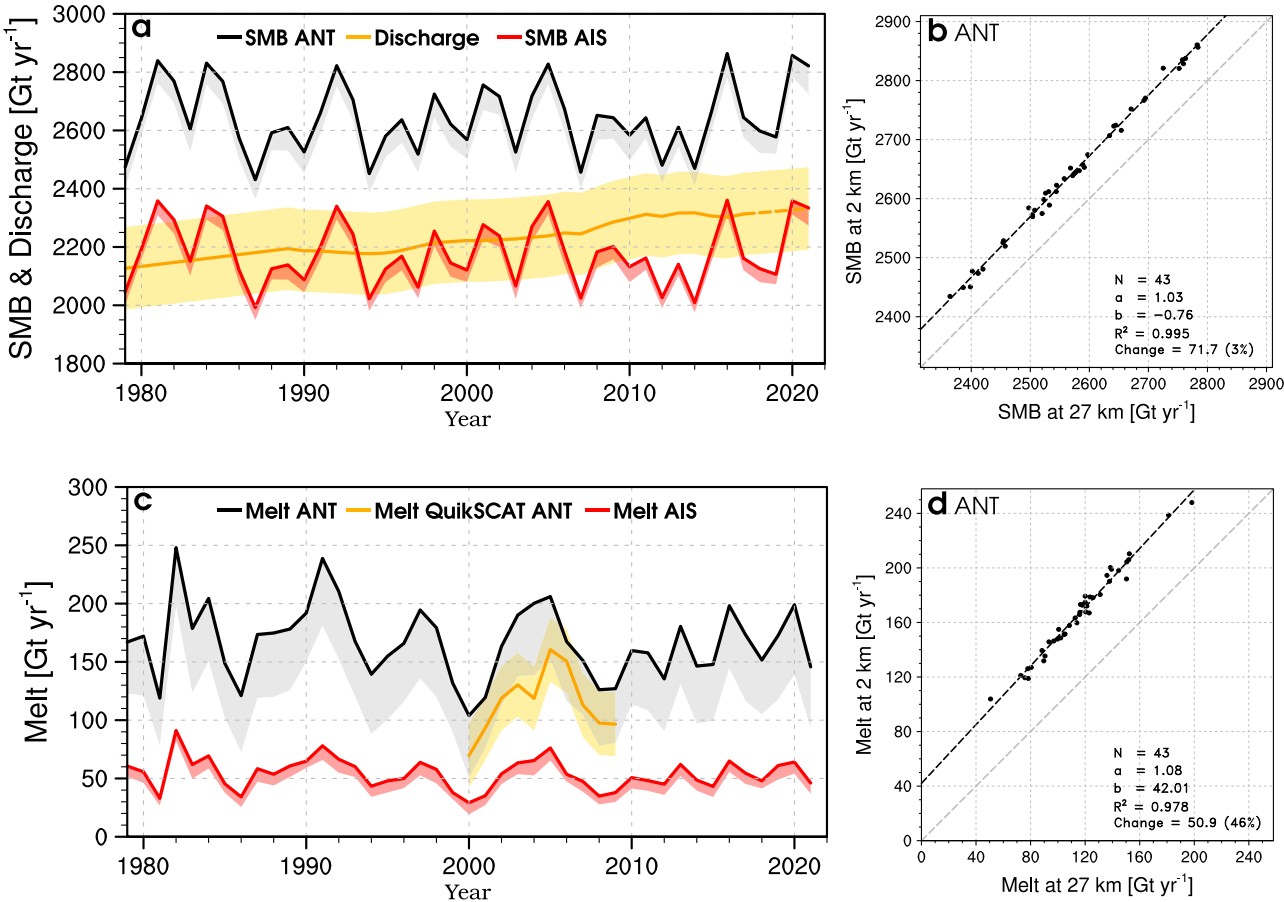

**Fig. 2 | Enhanced accumulation and surface melt at high resolution. (a)** Time series of annual surface mass balance (SMB) integrated over the grounded Antarctic ice sheet including Antarctic islands (AIS, red) and Antarctic-wide including the floating ice shelves (ANT, black) for the period 1979–2021. The red and black lines represent the 2 km product; the red and grey bands represent the change from the 27 km product. The yellow line and band represent the annual solid ice discharge and uncertainty from Rignot et al.[1]. As the solid ice discharge data set does not extend beyond 2017, it is linearly extrapolated thereafter and shown as a dashed line. **(b)** Cross model correlation between SMB at 2 km and 27 km, spatially integrated over the whole of Antarctica (ANT). **(c)** Time series of annual surface melt integrated over the grounded ice sheet (AIS, red) and Antarctic-wide (ANT, black) for the period 1979–2021. The red and black coloured lines represent the 2 km product, while the red and grey bands represent the change from the 27 km product. QuikSCAT melt and uncertainty (2000–2009)[25] integrated over the whole of Antarctica (ANT) are represented as orange line and band. Uncertainty in QuikSCAT is estimated as one standard deviation around the 2000–2009 mean. **(d)** Cross model correlation between melt at 2 km and 27 km, spatially integrated over the whole of Antarctica (ANT). In **(b)** and **(d)** relevant statistics including number of records ($N$), the slope ($a$) and intercept ($b$) of the regression line ($y = ax + b$), coordination coefficient ($R^2$), and mean model difference expressed as an absolute and relative (brackets) value are listed.

Using 14,292 multi-year in situ SMB measurements from the AntSMB data set[37] (white dots in Supplementary Fig. 2a), we show that the 2 km product improves on the original 27 km data ($R^2 = 0.83$ vs. 0.80), with both lower bias (−6.6 mm w.e.) and RMSE (−9.4 mm w.e.) (Supplementary Fig. 2b, c). Using a two-sample Kolmogorov-Smirnov (K-S) test on the model bias at both resolutions, we find that although small, these local SMB changes at 2 km are significant ($p$ value < 0.01). We integrate SMB over the whole of Antarctica (ANT), including the grounded AIS, adjacent Antarctic islands, and floating ice shelves. Accumulation in the 2 km product (black line in Fig. 2a) increases by 72 Gt year⁻¹ (3% for the period 1979–2021 in Fig. 2b) relative to the 27 km product (grey band in Fig. 2a). This difference falls well within the multi-model uncertainty of 266 Gt year⁻¹ estimated by Mottram et al.[5]. Elevation correction onto the 2 km grid accounts for 49 Gt year⁻¹ (2%) of the total SMB increase, whereas the remaining 23 Gt year⁻¹ (1%) are attributed to differences in ice mask extent (Supplementary Fig. 1c, d). Dividing Antarctica in sectors, we find that most of the accumulation increase originates from the Antarctic Peninsula (APIS) (31 Gt year⁻¹ in Supplementary Fig. 3a), followed by the floating ice shelves (27 Gt year⁻¹ in Supplementary Fig. 3d), and the East Antarctic ice sheet (EAIS) (16 Gt year⁻¹ in Supplementary Fig. 3c). The West Antarctic ice sheet

(WAIS), however, experiences a small and negligible accumulation decrease (2 Gt year⁻¹ in Supplementary Fig. 3b). Regional changes in area-integrated SMB at both resolutions are summarised in Supplementary Table 1.

**Contemporary mass change estimates**

Mass change (MB) of the grounded AIS can be quantified using the input-output method[2], i.e., the difference between modelled SMB (this study) and solid ice discharge (D)[1] for the period 1979–2021 (Fig. 2a). Since published D estimates do not extend beyond 2017, we extrapolate the time series to 2021 using a linear regression for 1979–2017 ($R^2 = 0.95$) (see "Methods"). In the past four decades (1979–2021), we show that modelled SMB at 2 km integrated over the grounded AIS, including Antarctic islands (blue region in the inset map of Fig. 3a), remained relatively constant at 2180 ± 107 Gt year⁻¹ of surface accumulation with an insignificant negative trend of −0.001 ± 0.660 Gt year⁻² ($p$ value > 0.01) (red line in Fig. 2a). In contrast, solid ice discharge (D) contributed an average mass loss of 2220 ± 142 Gt year⁻¹ for the period 1979–2017[1], with a significant positive trend of 4.9 ± 0.4 Gt year⁻² ($p$ value < 0.01) (orange line in Fig. 2a). For evaluation, we estimate monthly mass change at both spatial resolutions and compare

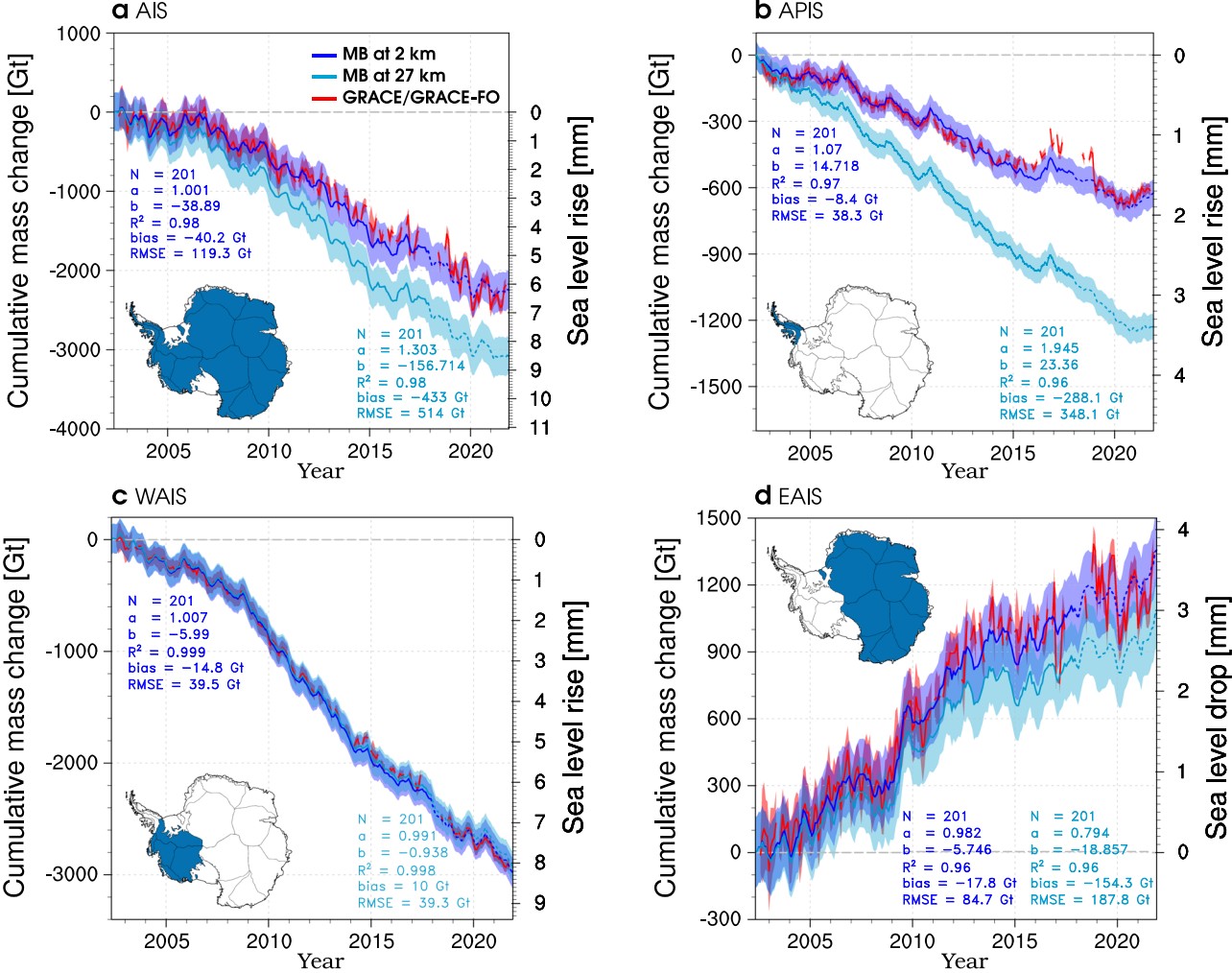

**Fig. 3 | Regional mass change since 2002.** Time series of monthly cumulative mass change (MB) for 2002–2021 at 2 km (blue line) and 27 km (cyan line) for (**a**) the grounded Antarctic ice sheet (AIS), (**b**) the Antarctic Peninsula (APIS), (**c**) the West Antarctic ice sheet (WAIS), (**d**) the East Antarctic ice sheet (EAIS) sectors. Monthly mass change is estimated as modelled surface mass balance (SMB) at 2 km or 27 km minus (regional) solid ice discharge from Rignot et al.[1]. As the solid ice discharge data set does not extend beyond 2017, it is linearly extrapolated thereafter. The resulting MB at both resolutions is shown as dashed lines after 2017. Regional mass change of grounded ice on surrounding Antarctic islands is included (see "Methods"). Mass change is converted into global sea level change assuming that 362 Gt of ice raises sea level by 1 mm. Inset maps show in blue the region of interest. Monthly mass change from GRACE/GRACE-FO is shown as red lines. Uncertainty estimates (coloured bands) are described in the "Methods". Relevant statistics including number of records ($N$), the slope ($a$) and intercept ($b$) of the regression line ($y = ax + b$), coordination coefficient ($R^2$), mean bias and root mean square error (RMSE) are listed.

the results with that of the Gravity Recovery and Climate Experiment (GRACE) Follow-On (FO) satellite products for the overlapping period 2002–2021 (see "Methods"). We show that Antarctic-wide mass change from the 2 km product (blue line in Fig. 3a) significantly improves on that from the 27 km product (cyan line), reducing both mean bias and RMSE by about 395 Gt relative to GRACE/GRACE-FO (red line). We also find significant improvements in the APIS (Fig. 3b) and the EAIS sectors (Fig. 3d). In the WAIS sector, the mean bias and RMSE are similar at both resolutions (Fig. 3c). Overall, the 2 km product agrees better with GRACE/GRACE-FO records both AIS-wide ($R^2 = 0.98$) and over its three sectors ($0.96 < R^2 < 0.99$).

Supplementary Fig. 4 extends our mass change estimates back to 1979. Using the 2 km product, we find that the grounded AIS remained in approximate mass balance until the mid-2000s (blue line in Supplementary Fig. 4a), with SMB mass gain ($2187 \pm 107$ Gt year$^{-1}$ for 1979–2005) compensating for D mass loss ($2187 \pm 142$ Gt year$^{-1}$) on average (Fig. 2a), though with large interannual variability. Thereafter, D ($2301 \pm 142$ Gt year$^{-1}$ for 2006–2021) persistently exceeds SMB ($2168 \pm 107$ Gt year$^{-1}$) (Fig. 2a), driving the recent mass loss

(Supplementary Fig. 4a). We estimate that the grounded AIS has lost $2272 \pm 243$ Gt of ice since 2002, contributing $6.3 \pm 0.7$ mm to global sea-level rise (Fig. 3a). The APIS and WAIS sectors (Fig. 3b, c), contributed $634 \pm 61$ Gt and $2930 \pm 132$ Gt to the total mass loss, respectively. In contrast, persistent mass gains in the EAIS sector since 2002 mitigate mass losses from the other two sectors by $1292 \pm 158$ Gt (Fig. 3d). The climatic signal of the EAIS mass gain remains actively debated[1–3,6,38]. Nevertheless, our downscaled (regional) SMB estimates are supported by good agreement with both in situ (Supplementary Fig. 2) and remote sensing records (Fig. 3d). Regional contributions to global sea level change since 2002 are listed in Supplementary Table 2.

## Increased surface melt rates at the grounding line
Among other processes[4], floating ice shelves are vulnerable to (local) high surface melt rates[17] that can result in meltwater ponding, trigger ice shelf destabilisation and rapidly accelerate solid ice discharge. It is therefore critical to accurately represent surface melt rates, notably near the grounding line that separates the grounded AIS from the floating ice shelves. Statistical downscaling to 2 km has a pronounced

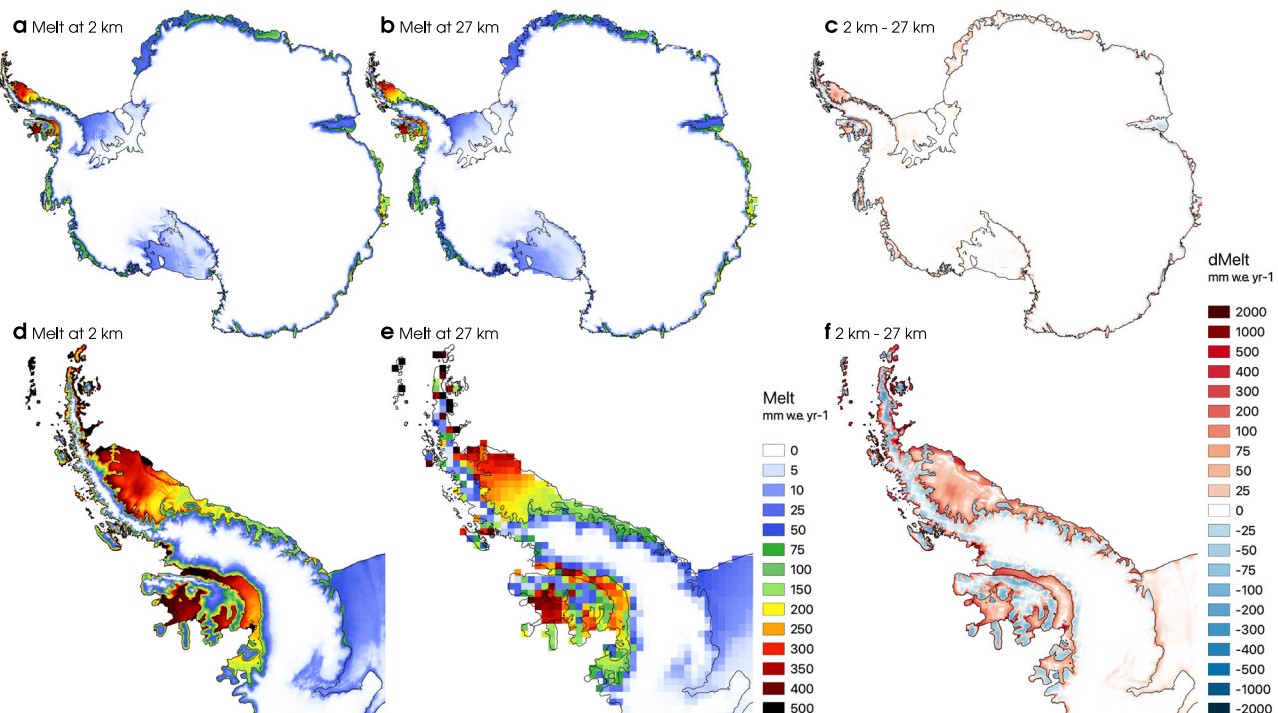

**Fig. 4 | High-resolution surface melt patterns in Antarctica.** Modelled mean annual Antarctic surface melt for the period 1979–2021 (**a**) statistically downscaled to 2 km resolution, (**b**) modelled by RACMO2.3p2 at native 27 km resolution. (**c**) Melt difference between the 2 km and 27 km products. (**d–f**) Same as (**a–c**) but for the Antarctic Peninsula.

effect on surface melt rate and distribution across Antarctica, especially over the ice shelves where melt peaks (Fig. 4a). Differences between modelled melt at both resolutions locally reaches up to 500 mm w.e. year⁻¹ (Fig. 4c). As for SMB, varying ice mask extent between both grids can lead to large differences, notably at the ice mask edge (Supplementary Fig. 1c, d). Overall, melt changes mirror the surface topography difference between the 2 km and 27 km grids, reaching up to 500 m (Supplementary Fig. 1a). As a general pattern, surface elevation in RACMO2.3p2 is overestimated over narrow, low-lying valleys and at the grounding line (Supplementary Fig. 1a). In contrast, surface elevation at 27 km is underestimated over the divide of the steep mountain ranges, including the Antarctic Peninsula, Queen Maud Land and the Transantarctic Mountains (Supplementary Fig. 1a). These mountain ranges with their networks of narrow peaks and valleys locally show large elevation biases (-500 m) between the 2 km and 27 km grid (Supplementary Fig. 1a). Small negative elevation differences are generally found over the relatively flat floating ice shelves (Supplementary Fig. 1a). Combined with the strong surface melt gradients found nearby the grounding lines, this leads to an overall melt increase that locally peaks over crevasses that are not resolved at 27 km (Fig. 4c). In contrast, the Amery ice shelf shows a small positive elevation difference at 2 km, leading to a local melt decrease compared to the 27 km grid (Fig. 4c and Supplementary Fig. 1a). Locally, the albedo correction implies that surface melt further increases in regions where blue ice is exposed. Lower elevations in the downscaled product result in higher melt rates relative to native RACMO2.3p2, and vice-versa (Fig. 4c). Such patterns are clearly visible in the APIS sector, with increased melt rates over low-lying valley glaciers, crevasses, and in the vicinity of the grounding line, while melt is reduced towards the elevated interior (Fig. 4f).

We evaluate modelled surface melt at 2 km and 27 km using 81 annual records from ten AWS (yellow stars in Supplementary Fig. 2a) spanning the period 1992–2018[24] (see "Methods"). Four sites are situated on the Larsen C ice shelf and six in Dronning Maud Land. We find

that the downscaled product shows better agreement with observations than the native 27 km product (Supplementary Fig. 2d, e), with reduced bias (−7.9 mm w.e.) and similar RMSE. Since the AWS network is spatially limited, we complement our evaluation with an Antarctic-wide point comparison between mean annual melt modelled on both grids and derived from QuikSCAT[25] at 4.45 km resolution, averaged for the period 2000–2009 (Supplementary Fig. 5a) (see "Methods"). We find that high melt rates are better resolved on the 2 km grid with reduced bias (−8.3 mm w.e.) and RMSE (−7.2 mm w.e.) (Supplementary Fig. 5b, c). Notably, the 2 km product improves upon the native RAC-MO2.3p2 product at capturing high melt rates measured in the vicinity of the grounding line (Supplementary Fig. 5d, e). Integrated over the whole of Antarctica (ANT), melt in the 2 km product increases by 51 Gt year⁻¹ (46% for 1979–2021 in Fig. 2d) relative to the 27 km product (black line and grey band in Fig. 2c). The elevation correction contributes 38 Gt year⁻¹ (34%) to the total surface melt increase (Supplementary Fig. 1a, b). This effect is particularly important near the grounding line where surface elevation is generally reduced at 2 km, and steep topographic gradients were not accurately captured at 27 km. Over low-lying ice shelves, the combined elevation difference and strong melt gradients locally enhance surface melt at 2 km. Spatial refinement of the ice mask from 27 km to 2 km contributes 10 Gt year⁻¹ (9%) to the total melt increase (Supplementary Fig. 1c, d), while the remaining 3 Gt year⁻¹ (3%) stem from albedo correction over blue ice areas. The downscaled product aligns with QuikSCAT estimates within uncertainties for the period 2000–2009 (orange line and band in Fig. 2c). Supplementary Fig. 3e–h shows similar results for individual sectors, highlighting a 42 Gt year⁻¹ (62%) increase in surface melt over floating ice shelves that mainly occurs along the grounding line (Fig. 4c). For the grounded AIS, melt and therewith the melt increase is smaller with 1 Gt year⁻¹ in the WAIS (39%), 2 Gt year⁻¹ in the APIS (9%), and 6 Gt year⁻¹ in the EAIS sector (45%) (Supplementary Fig. 3e, f). Regional changes in area-integrated surface melt at both resolutions are shown in Supplementary Table 3.

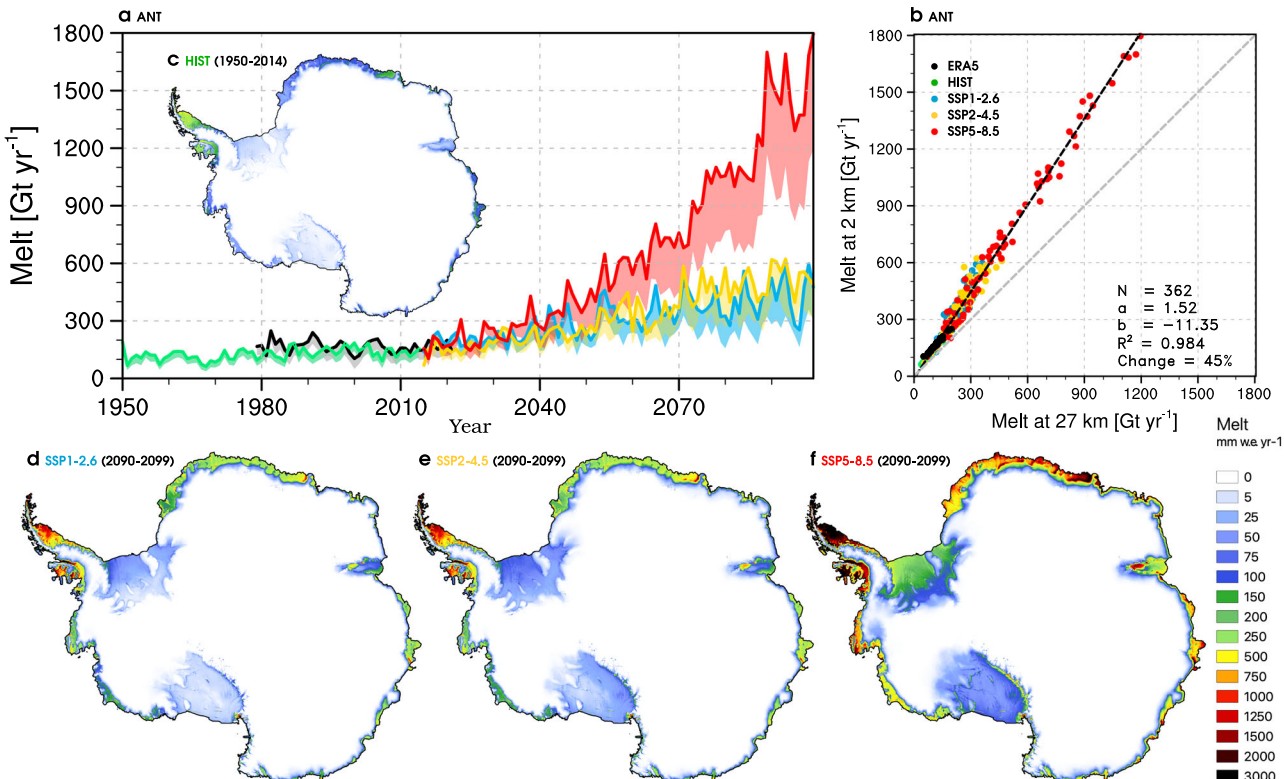

**Fig. 5 | Persistent surface melt increase at high-resolution until 2100. (a)** Time series of modelled annual surface melt integrated over the whole of Antarctica (ANT) for the period 1950–2099. Coloured lines represent surface melt statistically downscaled to 2 km from RACMO2.3p2 forced by ERA5 reanalysis (black, 1979–2021), and by CESM2 for the historical period (green, 1950–2014), and three climate projections under SSP1-2.6 (cyan), SSP2-4.5 (yellow) and SSP5-8.5 (red, 2015–2099). Coloured bands represent the change relative to the corresponding simulations from RACMO2.3p2 at 27 km resolution. **(b)** Cross model correlation of surface melt between the 2 km and 27 km products shown in (**a**). Mean annual surface melt from RACMO2.3p2 forced by (**c**) CESM2 in the historical period (1950–2014), and by three climate projections (2090–2099) under (**d**) SSP1-2.6, (**e**) SSP2-4.5 and (**f**) SSP5-8.5. In (**b**) relevant statistics including number of records (*N*), the slope (*a*) and intercept (*b*) of the regression line (*y = ax + b*), coordination coefficient (*R²*), and mean model difference expressed as a relative value are listed.

## Persistently higher surface melt rates until 2100

To explore whether the surface melt underestimate found in the contemporary RACMO2.3p2 simulation at 27 km persists in the future, we use one CESM2 historical reconstruction (HIST, 1950–2014) extended by three climate projections under a low (SSP1-2.6), moderate (SSP2-4.5) and high-end (SSP5-8.5) warming scenario (1950–2099). These simulations are first dynamically downscaled to 27 km using RACMO2.3p2, and further statistically downscaled onto the 2 km grid (see "Methods"). Figure 5a shows time series of Antarctic-wide (ANT) integrated melt at 2 km (dark coloured lines) and 27 km (lower margin of the light coloured bands) for the period 1950–2099. We first ascertain that melt in the HIST period (green, 1950–2014) aligns with the contemporary ERA5 simulation (black, 1979–2021). We find good agreement with an averaged melt at 2 km of $139 \pm 27$ Gt year⁻¹ in the HIST run compared to $167 \pm 33$ Gt year⁻¹ in the ERA5 simulation for the overlapping period 1979–2014 (Fig. 5a). In the period 1950–2014, Antarctic-wide (ANT) melt in the HIST run remained steady with a small positive trend of $0.6 \pm 0.4$ Gt year⁻². In line with previous studies[15,17,20], melt increases linearly under low ($3.5 \pm 0.6$ Gt year⁻² for 2015–2099) and moderate scenarios ($5.4 \pm 0.6$ Gt year⁻²), with a stronger acceleration under a high-end warming trajectory ($17.4 \pm 1.3$ Gt year⁻²). Figure 5b compares the Antarctic-wide (ANT) melt between both model resolutions for all warming scenarios. We find that, irrespective of the warming trajectory, a systematic melt underestimate of 45% persists in the 27 km product (Fig. 5b), corroborating our results for the contemporary simulation (Fig. 2d). Dividing Antarctica in sectors, Supplementary Fig. 6 shows that the relative melt increase over

floating ice shelves (61%) is larger than for individual AIS sectors (11–24%).

## Discussion

Integrated over the grounded AIS, statistical downscaling increases SMB by 2% (45 Gt year⁻¹ for 1979–2021 in Fig. 2a). Although small, this SMB change has a pronounced impact on (regional) mass change estimates (Fig. 3). Combined with observed solid ice discharge (D)[1], the modest increase in the 2 km SMB product is sufficient to reconcile modelled and remotely sensed (GRACE/GRACE-FO) mass change both AIS-wide and at the regional scale (Fig. 3b–d), without further adjustments. Previous work using SMB from RACMO2.3p2 at 27 km found good agreement between modelled and GRACE/GRACE-FO mass change, but only after applying regional corrections to the modelled SMB trends[6], of similar magnitude to those obtained through statistical downscaling (Supplementary Fig. 3a–c). Using the 2 km product, we show that AIS mass loss since 2002 contributed $6.3 \pm 0.7$ mm to global sea level rise in excellent agreement with GRACE/GRACE-FO (Fig. 3a), essentially reducing the AIS contribution by a third compared to the 27 km estimate ($8.6 \pm 0.7$ mm) (Supplementary Table 2). The difference between both resolutions primarily stems from too low accumulation at 27 km (Supplementary Fig. 3a, c), which result in mass loss overestimate in the APIS (1.7 mm SLE in Fig. 3b) and mass gain underestimate in the EAIS (0.8 mm SLE in Fig. 3d). In the WAIS sector, both products perform similarly though with slightly lower mass loss at 27 km (0.2 mm SLE in Fig. 3c), following an insignificant SMB decrease at 2 km resolution (Supplementary Fig. 3b). Extending mass change time series back to 1979, these resolution-driven differences

become even more pronounced (Supplementary Fig. 4). This high-lights the added value of statistical downscaling to improve our present-day reconstructions of mass change in Antarctica.

Although essential to predict the viability of floating ice shelves[17], Antarctic-wide (ANT) integrated surface melt estimates from current state-of-the-art regional climate models largely differ: 197 ± 43 Gt year$^{-1}$ in MAR3v12 at 35 km (1980–2021)[19,20], 116 ± 29 Gt year$^{-1}$ in RACMO2.3p2 at 27 km (1980–2021)[18], and 695 ± 132 Gt year$^{-1}$ in HIRHAM5 at 12.5 km resolution (1980–2017)[39]. Applying statistical downscaling to RAC-MO2.3p2 at 27 km, we show that melt at 2 km increases by 45% to reach 167 ± 32 Gt year$^{-1}$ (black line and grey band in Fig. 2c). Melt mostly increases along the grounding line, in good agreement with in situ AWS measurements (Supplementary Fig. 2d, e) and QuikSCAT records (Supplementary Fig. 5). We find that the higher surface melt rates persist in the future irrespective of the warming scenario, both Antarctic-wide and at the sector scale (Fig. 5a, b and Supplementary Fig. 6), suggesting higher future melt rates than previously estimated. While the magnitude of melt increase remains small in the present-day (51 Gt year$^{-1}$ for 1979–2021) (grey band in Fig. 5a), we predict that it could become over 3-fold (170 Gt year$^{-1}$) to over 10-fold (525 Gt year$^{-1}$) larger in 2090–2099 under SSP1-2.6 (cyan band) and SSP5-8.5 (red band). Most of the melt increase relative to present-day occurs along the grounding line fringing floating ice shelves, especially over Larsen C and Dronning Maud Land (Fig. 5c–f). In view of the rapid surface melt increase in the future (Fig. 5a), accurate representation of melt rates and patterns are crucial to identify the onset of potential ice shelf collapse and subsequent mass loss acceleration[16,17]. Statistical down-scaling, that captures daily surface melt and SMB in unprecedented spatial detail, is an efficient tool to explore future ice shelf viability and quantify the AIS contribution to global sea level rise.

## Methods

### Regional Atmospheric Climate Model
The polar (p) Regional Atmospheric Climate Model version 2.3p2 (RACMO2.3p2) incorporates the dynamical core of the High Resolution Limited Area Model (HIRLAM)[40] and the physics package cycle CY33r1 of the European Centre for Medium-Range Weather Forecasts-Integrated Forecast System (ECMWF-IFS)[41]. The model is specifically adapted to represent surface processes of polar ice sheets and ice caps including the Greenland ice sheet[30], Canadian Arctic[33], Svalbard[34], Iceland[35], and Antarctica[18]. The model incorporates a 40-layer snow module (up to 100 m depth) simulating melt, percolation and retention into firn and subsequent surface runoff[42]. The model represents dry-snow densification[43], drifting snow erosion[44], and snow albedo based on grain size, cloud optical thickness, solar zenith angle, and impurity content[45]. Here we use the benchmark RACMO2.3p2[18] model at 27 km spatial resolution to dynamically downscale the latest ERA5 climate reanalysis (1979–2021)[26] from the European Centre for Medium-Range Weather Forecasts (ECMWF). This reference run is further complemented by one historical simulation (1950–2014) and three climate projections from the Community Earth System Model (CESM2) under a low-end (SSP1-2.6), moderate (SSP2-4.5) and high-end (SSP5-8.5) Shared Socioeconomic Pathways (SSP) emission scenarios (2015–2099), dynamically downscaled to 27 km. RACMO2.3p2 is forced by ERA5 reanalysis and CESM2 climate outputs within a 24-grid-cell-wide relaxation zone at the lateral model boundaries. Forcing consists of temperature, pressure, specific humidity, wind speed and direction being prescribed at the 40 model atmospheric levels every 3 and 6 h, for the ERA5 and CESM2-forced setting respectively. Upper atmospheric relaxation is active[46]. Sea ice extent and sea surface temperature are also prescribed from the ERA5 reanalysis or CESM2 outputs on a 3 or 6-h basis. Firn is initialised in January 1979 (ERA5) and 1950 (CESM2) using snow temperature and density profiles from the offline Institute for Marine and Atmospheric research Utrecht-Firn Densification Model (IMAU-FDM)[43]. Ice albedo is prescribed as a

constant field in space and time at 0.55. Surface topography is pre-scribed from the 1 km digital elevation model of Bamber et al.[47], down-sampled to 27 km. In the absence of interactive ice dynamics in RAC-MO2.3p2, ice mask and topography are kept fixed. The RACMO2.3p2 model settings are discussed and evaluated in van Wessem et al.[18].

### Community Earth System Model
The Community Earth System Model version 2.1 (CESM2.1)[27] is an earth system model from the Climate Model Comparison Project phase 6 (CMIP6) that simulates interactions between atmosphere-land-ocean systems on the global scale. The model includes the Community Atmosphere Model version 6 (CAM6)[48], the Parallel Ocean Program model version 2.1 (POP2.1)[49], the Los Alamos National Laboratory Sea Ice Model version 5.1 (CICE5.1)[50], the Community Land Model version 5 (CLM5)[51], and the Community Ice Sheet Model version 2.1 (CISM2.1)[52] to simulate interactions between atmosphere-ocean-land and sea ice dynamics as well as snow/ice surface processes in a fully-coupled fashion. Here all components are active except for land-ice dynamics, i.e., excluding basal melting, calving, collapse of ice shelves and subsequent ice sheet thickening/thinning and advance/retreat. We use outputs from the CMIP6 CESM2 model for one historical reconstruction (HIST; 1950–2014), and three climate projections (SSP1-2.6, SSP2-4.5 and SSP5-8.5; 2015–2099)[53] at a spatial resolution of 0.9 × 1.25°. In CESM2, CAM6 and CLM5 exclusively prescribe land use changes; atmospheric greenhouse gas and aerosol emissions are derived from the selected SSP warming scenarios[54]. CLM5 has been adapted to simulate runoff, melt, firn retention and refreezing in a 10 m w.e. snowpack[55]. Model settings are discussed and evaluated in Dunmire et al.[21].

### Statistical downscaling
Following Noël et al.[29], the RACMO2.3p2 contemporary climate (1979–2021, ERA5) and projections (1950–2099, CESM2) are statistically downscaled from the native model resolution of 27 km to an ice mask and topography defined on a 2 km grid. A 2 km grid was selected as a trade-off between computational efficiency and high spatial resolution, while maintaining manageable file size. The ice mask combines the grounded AIS mask from the Ice Sheet Mass Balance Inter-comparison Exercise-2 (IMBIE-2)[2], the refined floating ice shelf mask from the Antarctic Bedrock Mapping version 2 (Bedmap2)[56], and sub-Antarctic glaciers from the Randolph Glacier Inventory version 6 (RGIv6)[57] down-sampled to a 2 km grid. Surface topography at 2 km is derived from the Reference Elevation Model of Antarctica (REMA)[28] digital elevation model at 200 m resolution. The downscaling procedure corrects individual SMB components for elevation on the 2 km topography, using daily-specific vertical gradients estimated on the RACMO2.3p2 grid at 27 km. SMB components ($X$ in Eq. (1)) include total precipitation (PR), total sublimation (SU), snow drift erosion (ER), total melt (ME), and runoff (RU). Vertical gradients are estimated as linear regressions using at least six grid-cells, i.e., the current one and five (or more) adjacent pixels. To obtain realistic local estimates, the regression slope ($a$) is applied to the current grid-cell to compute an intercept ($b$), i.e., value at sea level. These two regression coefficients are bi-linearly interpolated from the 27 km grid onto the 2 km one, and applied to the high-resolution topography at 2 km ($h$) as:

$$X_{2km} = a_{2km} \times h_{2km} + b_{2km} \qquad (1)$$

Melt and runoff are further corrected for surface albedo in regions exposing blue ice or darker bare ice mixed with rocks (albedo < 0.55) in summer, that are unresolved in RACMO2.3p2 at 27 km. To that end, we use a 2 km MODIS 16-day product averaged for the period 2000–2021. The albedo correction accounts for solar zenith angle, surface slope and aspect as discussed in Noël et al.[29]. Since runoff is spatially limited in Antarctica, i.e., surface melt is mostly retained or refrozen in firn, we

bi-linearly interpolate runoff in space onto the 2 km grid for days with insufficient number of grid-cells experiencing runoff (<6 adjacent grid-cells). This resolution issue becomes less pronounced as the runoff zone expands following atmospheric warming, notably in the three SSP scenarios. Statistical downscaling of total precipitation is motivated by the fact that the native RACMO2.3p2 underestimates orographic precipitation, notably on the steep and rugged terrain of the Antarctic Peninsula. In addition, daily snowfall fraction ($SF_{frac}$), i.e., the fraction of snowfall on total precipitation, is statistically downscaled to 2 km following Huai et al.[58]. Snowfall is thus estimated as:

$$SF_{2km} = PR_{2km} \times SF_{frac2km} \qquad (2)$$

Rainfall is estimated as a residual as:

$$RA_{2km} = PR_{2km} - SF_{2km} \qquad (3)$$

SMB is reconstructed using individual components statistically downscaled to 2 km as:

$$SMB_{2km} = PR_{2km} - RU_{2km} - SU_{2km} - ER_{2km} \qquad (4)$$

and refreezing (RF) is estimated as a residual following:

$$RF_{2km} = ME_{2km} + RA_{2km} - RU_{2km} \qquad (5)$$

The above statistical downscaling procedure is not mass conservative, enabling mass flux corrections that significantly improve the representation of individual SMB components relative to the native RACMO2.3p2 outputs. For a detailed description of the statistical downscaling technique, we refer to Noël et al.[29].

## Limitations and uncertainties

The statistical downscaling technique used in this study builds on previous RACMO2-based downscaling efforts over glaciated surfaces in Greenland[29–32], the Canadian Arctic[33], Svalbard[34], and Iceland[35]. For all these land ice masses, the downscaled product accurately represents the spatial and temporal SMB variations, as demonstrated by high and improved correlations with in situ and remotely sensed observations. The statistical downscaling technique uses elevation and albedo as primary downscaling predictors and applies the downscaling at very high (daily) temporal resolution. Three factors explain the robustness of the downscaling procedure. First, the spatial variability of individual SMB components shows strong average correlation with elevation, also in Antarctica (Supplementary Fig. 7). Second, local albedo-driven surface melt anomalies are captured by implementing a spatial correction for ice albedo in the downscaling procedure, which accounts for solar zenith angle, slope, and aspect of the underlying terrain as detailed in Noël et al.[29]. For instance, the albedo correction locally enhances surface melt over darker blue ice areas. As blue ice currently remains spatially limited in Antarctica, albedo correction only contributes -1/15th (3%) to the total surface melt increase obtained after downscaling (+46%). Finally, statistical downscaling estimates elevation gradients locally (using the current and at least five adjacent grid-cells) and at high temporal resolution (daily). This preserves the modelled spatial patterns and synoptic timescales enabling to capture melt and accumulation spatiotemporal variability due to local surface slope, distance to the coast, and atmospheric circulation direction and strength affecting e.g., foehn events. Sources of uncertainty that persist in the current statistical downscaling routine stem from local biases in: (1) the modelled elevation gradients (RACMO2.3p2[17,18] or CESM2[21]) notably in regions characterised by steep slopes and sharp gradients, (2) the high-resolution surface topography (REMA)[28] used for the elevation correction, (3) the ice sheet (IMBIE-2)[2], glaciers (RGIv6)[57] and ice shelf masks (Bedmap2)[56]

used for spatial integration, and (4) the satellite albedo product (MODIS) used for ice albedo correction.

RACMO2.3p2[17,18] and none of the CESM2 scenarios[21] currently available implement the effect of ice dynamics in Antarctica, i.e., fixed present-day conditions were prescribed in our future projections. Under climate warming, the AIS and floating ice shelves are expected to retreat, calve and/or thicken/thin, which would affect the circulation and associated precipitation distribution as well as melt patterns. For instance, marginal ice thinning could trigger melt-elevation feedback, further enhancing the projected surface melt rates, while snowfall-driven ice thickening could reduce melt. These competing mechanisms will be investigated by statistically downscaling future melt projections from forthcoming fully-coupled earth system models that include Antarctic ice dynamics when these become available, as was previously done for the Greenland ice sheet[35]. In addition, we prescribe the present-day MODIS blue ice albedo with a base value of 0.55 when firn retreats to expose blue ice both in RACMO2.3p2 at 27 km and at 2 km. This approach remains valid in the absence of future surface darkening due to biological growth (algae)[59] or impurity deposition (black carbon, dust, ashes)[60] as e.g., observed in Greenland. To our knowledge, ice darkening is not yet ongoing in Antarctica. In view of the above, our projected, statistically downscaled surface melt rates should be interpreted as conservative estimates.

## SMB and mass change observations

For point SMB evaluation, we use in situ measurements from the AntSMB data set[37] that compiles multi-year records from stakes, snow pits, ice cores, ultrasonic sounders and ground penetrating radar covering the past 1000 years (white dots in Supplementary Fig. 2a). Here, we discarded records that were located outside the ice mask at 2 km. For a meaningful comparison with observations that did not overlap with our benchmark ERA5-forced RACMO2.3p2 simulation, we averaged modelled SMB at 2 km and 27 km over the period 1979–2021. To reduce the noise in our evaluation, we spatially averaged all point measurements that fell within a same 2 km grid-cell in the downscaled product. In total, 14,292 multi-annual records were compared to RACMO2.3p2 at 27 km and statistically downscaled to 2 km using the closest model grid-cell (blue dots in Supplementary Fig. 2b, c). For completeness, we also show the full data set as grey dots (274,154 records).

For large-scale SMB evaluation, we compare monthly mass change from the Gravity Recovery and Climate Experiment (GRACE, 2002–2018) and Follow-On mission (GRACE-FO, 2018–2021) (red lines in Fig. 3 and Supplementary Fig. 4) with combined sector integrated SMB statistically downscaled to 2 km (1979–2021) and corresponding solid ice discharge from Rignot et al.[1] (1979–2017). As the solid ice discharge data set does not extend beyond 2017, we apply a linear regression on the 1979–2017 period to extrapolate the time series until 2021. Although potentially missing discharge events after 2017, using a linear regression is most appropriate provided its high statistical significance ($R^2 = 0.95$, $p$ value < 0.01), and the fact that AIS mass change variability is primarily driven by SMB processes[6]. Sectors include the APIS, WAIS, EAIS and surrounding Antarctic islands, i.e., detached from the grounded AIS, as well as the whole grounded AIS. For a meaningful comparison with remote sensing records, i.e., that do not discriminate surrounding islands from the AIS sectors, we partition the islands' solid ice discharge and associated uncertainty ($D = 77 \pm 5$ Gt year$^{-1}$ in 1979–2017) into the three AIS sectors. Best agreement with remote sensing is found by attributing 20% of the flux to the APIS, 10% to the EAIS and the remainder to the WAIS sector. Note that these adjustments are small relative to the total flux in each sector (1–7% of the regional total), which only marginally affects our mass change estimates. Uncertainty in regional solid ice discharge is derived from Rignot et al.[1]. Uncertainty in modelled SMB at 2 km and 27 km is estimated as one standard deviation around the 1979–2021 mean. To

estimate modelled mass change uncertainty, we sum the corresponding uncertainty from SMB and D.

Mass anomalies of the AIS and its three sectors are derived from monthly GRACE and GRACE-FO spherical harmonic gravity field solutions from four processing centres, i.e., CSR RL06 from the Center for Space Research[61], JPL RL06 from the Jet Propulsion Laboratory[62], GFZ RL06 from the German Research Centre for Geosciences[63] and ITSG-Grace2018 from TU Graz[64]. We added degree-1 spherical harmonics as in Swenson et al.[65], and replaced the poorly resolved $C_{20}$ and $C_{30}$ coefficients by values estimated from satellite laser ranging, following the recommendations of the processing centres. Mass variations are estimated in 27 Antarctic basins following the method described in van Wessem et al.[18], for each of the four GRACE/GRACE-FO data sets. The resulting time series are then combined using error-weighted averaging as in Wouters et al.[66]. The correction for glacial isostatic adjustment is based on the model by Ivins and James[67].

### Surface melt observations

For local melt evaluation, we use 81 annual melt records from ten AWS covering the period 1992–2018[24]. Six stations are installed in Dronning Maud Land and four on the Larsen C ice shelf. We use the closest model grid-cell at 2 km and 27 km resolution for comparison with observations (Supplementary Fig. 2d, e). We also perform an Antarctic-wide point comparison between each grid-cell of the 2 km and 27 km products with QuikSCAT data at 4.45 km resolution, averaged for the period 2000–2009 (Supplementary Fig. 5a). To that end, native RAC-MO2.3p2 and QuikSCAT data are interpolated onto the 2 km grid to enable a grid-cell to grid-cell comparison. To remove spatial noise due to the different grid resolutions used, we apply an average filter of seven grid-cells (three pixels around the current one) to both high-resolution QuikSCAT and 2 km downscaled melt data sets before performing the point comparison (Supplementary Fig. 5b–e). These point comparisons are performed both Antarctic-wide and in the vicinity of the grounding line, i.e., within a RACMO2.3p2-derived grounding line contour at 27 km. We also compare Antarctic-wide (ANT) integrated melt from the 2 km (black line in Fig. 2c) and 27 km products (grey band in Fig. 2) with that of QuikSCAT (2000–2009, orange line in Fig. 2c)[25]. Uncertainty in QuikSCAT melt is estimated as one standard deviation around the 2000–2009 mean (orange band in Fig. 2c).

### Data availability

The statistically downscaled annual Antarctic-wide SMB (1979–2021) and surface melt (1950–2099) data sets at 2 km presented in this study have been deposited on Zenodo [https://zenodo.org/records/10007855][68]. Larger files including the gridded, daily downscaled SMB (components) data sets from the ERA-forced RACMO2.3p2 simulation, and the CESM2-forced RACMO2.3p2 projections under a low-end SSP1-2.6, moderate SSP2-4.5 and high-end SSP5-8.5 warming scenario are freely available from the authors upon request and without conditions (contact: bnoel@uliege.be).

### Code availability

The statistical downscaling technique is presented in Noël et al.[29]. The regional climate model RACMO2.3p2 is presented in van Wessem et al.[18]. The earth system model CESM2 is presented in Dunmire et al.[21]. The CESM2-forced RACMO2.3p2 data are presented in van Wessem et al.[17].

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

## Acknowledgements

B.N. was funded by the Fonds de la Recherche Scientifique de Belgique (F.R.S.-FNRS). This publication was supported by the project PROTECT funded by the European Union's Horizon 2020 research and innovation programme under grant agreement No. 869304, PROTECT contribution number 77. M.R.v.d.B. acknowledges support from the Netherlands Earth System Science Centre (NESSC).

## Author contributions

B.N. designed the study, prepared the manuscript, and statistically downscaled the presented data sets to 2 km. J.M.v.W. conducted the RACMO2.3p2 simulations at 27 km. B.N. and M.R.v.d.B. analysed the data. B.W. provided mass change records from GRACE/GRACE-FO. L.T. provided the QuikSCAT data set. S.L. helped to prepare the MODIS albedo time series. B.N., J.M.v.W., B.W., L.T., S.L. and M.R.v.d.B. commented on the manuscript.

## Competing interests

The authors declare no competing interests.
