## [Peer Review File · Nature Communications]

Higher Antarctic ice sheet accumulation and surface melt rates revealed at 2 km resolutionREVIEWER COMMENTS

Reviewer #1 (Remarks to the Author):

Please see uploaded file. 
Reviewer #2 (Remarks to the Author):

Review of Noël et al., “Higher Antarctic ice sheet accumulation and melt rates revealed at 2 km resolution”

What are the noteworthy results?

Statistical downscaling at 2km resolution is an effective tool to produce estimates of Antarctic surface mass balance (SMB) and melt in the present-day and future. It improves upon models with coarser spatial resolution by better capturing the topography of the Antarctic ice sheet, particularly on the Antarctic Peninsula. Over the historical period, statistical downscaling at 2km increases SMB by 2% over the Antarctic Ice Sheet compared to 27km resolution RACMO2.3p2. When this downscaled SMB is combined with discharge using the input-output method from 1979-2021, the calculated mass balance is much closer to remote sensing observations of mass balance from GRACE. Also in the present day, it is found that RACMO2.3p2 underestimates surface melting near the grounding line, due to an overestimate of elevation. In low, middle, and high-end projections for the future climate, the systematic underestimate of melt persists in the 27km product compared to the 2km.

Will the work be of significance to the field and related fields? How does it compare to the established literature?

The work is original and highlights the important impacts of elevation on modeled SMB and surface melt. Most climate models have a spatial resolution on the order of 10–100km, and this study emphasizes that these grids cannot resolve the complex topography of the Antarctic Ice Sheet, building on Lenaerts et al. (2019). This leads to an underestimation of accumulation on the windward side of mountainous regions and an underestimation of melt near the grounding line. The paper contextualizes the research gap by discussing results from Mottram et al. (2020) and Hansen et al. (2022), which show large variation in annual SMB estimates among multiple regional climate models linked to their grid resolutions. It builds on Dunmire et al. (2022), which evaluates SMB and melt in CESM2 in the historical period and examined future changes under three projected warming scenarios, providing important context for this work. Both Dunmire et al. (2022) and Kittel et al. (2021) highlight the contrast between melt and SMB on the grounded Antarctic Ice Sheet vs ice shelves in future climate scenarios, and this work expands on these studies by closely examining melt patterns along the grounding line at 2km resolution. The results are in agreement with previous studies on regional present and future SMB and melt in Antarctica, for example Donat-Magnin et al. (2021). While prior studies like Gilbert and Kittel (2021) have quantified Antarctic-wide future melt and SMB, this work does so at unprecedented spatial detail, enabling the authors to highlight the topographical effects missed in coarser resolution grids.

Does the work support the conclusions and claims, or is additional evidence needed?

The paper is clear regarding its scope and aims, explicitly stating that the primary purpose of statistical downscaling is to correct SMB components for elevation biases between the coarse topography of RACMO2.3p2 (27km) and the 2km REMA topography. Strong evidence is provided to support the claims, including the statistical significance of results comparing the 27km product to the 2km product, uncertainty metrics for SMB and discharge, and thorough comparisons to multiple observational datasets. The results will serve as a step forward in moving the field towards using Antarctic SMB products with finer spatial resolutions to examine the past, present, and future climate.

Are there any flaws in the data analysis, interpretation, and conclusions? Do these prohibit publication or require revision?

The authors present context for uncertainties in modelled SMB among multiple regional climate models in the introduction, and thoroughly compare modelled SMB and melt to observations during the historical period in the results. However, I think this paper would benefit from additional discussion on the uncertainties and biases that can be introduced or enhanced by the statistical downscaling method, in reference to existing biases in RACMO2.3p2, and how much these may influence the result – as well as in the future simulations, which rely on CESM2 and then RACMO2.3p2 prior to the downscaling. Additionally, the methods describe that the land-ice dynamics component of CESM2 is not active, including changes in ice sheet thickness. This implies that SMB- and melt-elevation feedbacks due to dynamical changes the ice sheet are not accounted for in the future scenarios. The authors showed evidence of the critical importance of elevation in producing snowfall and melt estimates for Antarctica, especially the Antarctic Peninsula. If the future projections do not account for changing ice sheet elevations in a warming climate, how might that impact the projected melt scenarios presented in the results?

Regarding the linear regression used to extend the discharge time series to 2021 (P3 L100), is there a precedent for this method? From Figures 2a and 3, it is clear that extending the discharge time series provides more data for comparison to GRACE-observed mass change, and that SMB is the primary driver for short-term variability in the mass balance during the 2018-2021 period, while discharge experiences a more linear increase. However, because the discharge and thus the mass change data are synthetic beyond 2017, I think it would be important to indicate this in the figures. I would strongly recommend adapting the line style, changing the shading, or adding a vertical line in the timeseries to indicate the period when discharge data are extrapolated.

Neither of these points prohibits publication. I would recommend minor revisions to address the uncertainty, the presentation of discharge extrapolated data, and minor comments below.

Is the methodology sound? Does the work meet the expected standards in your field? Is there enough detail provided in the methods for the work to be reproduced?

The methodology is sound and meets the expected standards in my field. The methods section is detailed, enabling reproducibility of the work regarding the use of RACMO2.3p2, CESM2, and

observations from the AntSMB data set, GRACE, and melt records from AWS. Please see limitations below regarding the statistical downscaling method.

Limitations

My primary limitation as a reviewer is that I do not have expertise in statistical downscaling. I cannot comment on the approach nor the reproducibility of that part of the method.

Minor Comments

P1 L4: “Surface processes are thus essential to quantify AIS mass change” groups ice shelf viability in with mass change; I would recommend either mentioning that ice shelf collapse can accelerate discharge or more distinctly separate ice shelf viability from mass changes.

P1 L12: It looks like the absolute melt increase is much greater in the high-end warming scenario than the low and middle scenarios (particularly on the Antarctic Peninsula and ice shelves in Supplementary Figure 6), so perhaps it would be better to use “in all scenarios” instead of “irrespective”.

P2 L41: “general melt underestimation in a regional climate model” – since multiple regional climate models are mentioned earlier in the paragraph, specificity would improve clarity here. Does this refer to RACMO2.3p1 and RACMO2.3p2?

P3 L66 and Figure 1f: It would be helpful if the difference in the 2km and 27km ice masks at the margins could be displayed in Figure 1f (as an outline of the 27km ice mask, or hatching, etc.), especially since the effect is highlighted in the text. Is it clear to see the resolution difference in Figure 1e, but it is less clear in the difference plot between 2km and 27km.

P3 L70 and Supplementary Figure 1: “2km – 27km” could be added at the top of each plot in Supplementary Figure 1, like in Figures 1c and 1f, to improve clarity.

P3 L83: Including both the definition of ANT and the results describing an increase in SMB from the 27km product to the 2km product forms a very long sentence here – would recommend simplifying or splitting in two.

P3 L85 and Figure 2: I find the colors used in Figures 2a and 2c slightly confusing, because almost the same 3 colors are used in each figure to represent different things. For example, to me it is non-intuitive that SMB AIS is portrayed in red in 2a and Melt AIS is portrayed in red in 2c, while at the same time SMB ANT is in black in 2a and Melt ANT is in yellow in 2c. It would be helpful if the authors could explain the reasoning for these color choices in a response or modify the colors to improve the consistency of color choices between the two figures.

P5 L143 and Figure 4f: As mentioned previously regarding Figure 1f, it would also be helpful if the difference in the 2km and 27km ice masks at the margins could be displayed in Figure 4f.

P5 L159 and Supplementary Figure 2a: I count 10 yellow stars in the figure (6 in DML), while 9 are mentioned in the text. Also, the stars are difficult to spot in the figure, can you increase their size and/or change the color to make them stand out better?

P5 L178: “increase in melt over floating ice shelves”

P12 L401: “(orange band in Figure 2d)”, “(black line in Figure 2d)”, and “(grey band in Figure 2d)” – should this be referring to 2c?

References

Cited in the paper and in the reviewer comments:

Dunmire et al. (2022)

Hansen et al. (2022)

Lenaerts et al. (2019)

Mottram et al. (2020)

Additional references:

Donat-Magnin, Jourdain, Kittel, Agosta, Amory, Gallée, Krinner, and Chekki (2021). “Future surface mass balance and surface melt in the Amundsen sector of the West Antarctic Ice Sheet”, *The Cryosphere*, doi: 10.5194/tc-15-571-2021

Gilbert and Kittel (2021). “Surface Melt and Runoff on Antarctic Ice Shelves at 1.5deg C, 2deg C, and 4deg C of Future Warming”, *Geophysical Research Letters*, doi: 10.1029/2020GL091733

Kittel, Amory, Agosta, Jourdain, Hofer, Delhasse, Doutreloup, Huot, Lang, Fichfet, and Fettweis (2021). “Diverging future surface mass balance between the Antarctic ice shelves and grounded ice sheet”, *The Cryosphere*, doi: 10.5194/tc-15-1215-2021

Reviewer #3 (Remarks to the Author): 
This is a very clear and well written manuscript describing a new downscaling of SMB in Antarctica using higher resolution than existing current state of the art downscaled products. The implications of higher resolution downscaling are well described and are important. The authors describe in particular differences that appear in the total surface mass balance integrated over Antarctica, and also differences locally, in particular in the Antarctic Peninsula where topographic gradients are most important. In addition, the authors demonstrate clearly how the new downscaling is able to explain offsets in previous downscaling between remote sensing data and reconstruction mass balance time series. This was particularly important, as a simple comparative study between low and higher-order resolution would have been difficult to validate. The implications were also

clearly explained in terms of projections and how this could impact the way surface melt-rates should be computed for increasingly warm scenarios at 2100 and beyond. I was very intrigued by the impact at the grounding line in particular, though I thought the authors put too strong an emphasis on hydrofracturing when describing the implications for ice shelf stability. More could be said about the impact on runoff in particular which has been found to be significantly disruptive to regional ocean circulation, a controlling factor of basal melt rates, another key component of the stability of ice shelves. I suggested changes to the wording in more details (see attached commented pdf). A little bit more effort should also be carried out in terms of anchoring this study against other existing downscaling efforts, maybe for example drawing from studies done at lower resolution for Greenland? Not being an SMB expert, I would like to understand how this anchored into the existing literature.

Overall, I think this is a worthy scientific addition to our knowledge, and should be shared with the Cryospheric community, I therefore recommend it be published following minor revisions to address in particular the way the study is introduced and anchored in the existing literature.

Response letter: Higher Antarctic ice sheet accumulation and surface melt rates revealed at 2 km resolution.

Dear Reviewers, we would like to thank you for your positive and constructive comments, and for giving us the opportunity to revise our manuscript. Below you will find our point answers in blue, and textual changes in red.

Reviewer #1 (Remarks to the Author):

General comments

In this clearly-written paper, the authors present estimates of surface mass balance (SMB) and surface melt over Antarctica derived from simulations using the RACMO2 regional climate model run at 27 km resolution which were statistically downscaled to an unprecedented 2 km resolution. Results are presented both for the present day (1979-2021) and for three future climate (to 2100) scenarios. For present-day climate, the downscaled simulations exhibit a small (3%) increase in Antarctic-wide SMB that, for the first time, brings model SMB estimates into agreement with Antarctic Ice Sheet mass changes derived from GRACE satellite gravimetric measurements. Downscaled surface melt is also in better agreement with satellite-derived melt than is melt from the 27 km simulation. The 2 km downscaled future climate scenarios show that the underestimate of melt in the 27 km present-day simulation persists into the future, independent of scenario.

These are important results that will help to reduce uncertainties in estimates of present-day SMB over Antarctica and its likely trajectory under future climate scenarios. As changes in Antarctic SMB will have an increasingly important impact on global sea level rise, this is a topic of high socioeconomic importance and I would expect to see this paper prominently cited in the next IPCC report. I strongly recommend its publication in Nature Communications subject to attention to the points listed below. Thank you!

Specific Comments

1. The downscaling approach used is based on the assumption that elevation is the most important control on the variables of interest to calculating SMB, so variables from the 27 km model can be downscaled to the 2 km grid by deriving local regression relationships of the variable as a function of elevation from the 27 km model data and then using these, together with 2 km elevation data from REMA, to evaluate the variable on the 2 km grid. This is a fairly standard approach to downscaling but it's not perfect – there will be some regions where parameters other than elevation (e.g. distance from the coast, terrain slope, etc.) will be important. I'm not suggesting that you should include all of these in your study but it might be worth including a few words on why you believe that elevation is probably the strongest control on the relevant variables over most of Antarctica, while noting that other controls may be important in some limited regions. This is a good point. The revised manuscript now includes new Supplementary Figure 7 and subsection *Limitations and uncertainties* in the Methods. This new subsection now 1) better supports our choices for elevation as main statistical downscaling predictor, 2) discusses the benefits of estimating elevation gradients at high spatial and temporal resolution in the downscaling approach, and 3) lists remaining sources of uncertainties, as follows: "The statistical downscaling technique used in this study builds on previous RACMO2-based downscaling efforts over glaciated surfaces in Greenland²⁹⁻³², the Canadian Arctic³³, Svalbard³⁴, and Iceland³⁵. For all these land ice masses, the downscaled product accurately represents the spatial and temporal SMB variations, as demonstrated by high and improved correlations with in-situ and remotely sensed observations. The statistical downscaling technique uses elevation and albedo as primary downscaling predictors and applies the downscaling at very high (daily) temporal resolution. Three factors explain the robustness of the downscaling procedure. First, the spatial variability of individual SMB components shows strong average correlation with elevation, also in Antarctica (Supplementary Fig. 7). Second, local albedo-driven surface melt anomalies are captured by implementing a spatial correction for ice albedo in the downscaling procedure, which accounts for solar zenith angle, slope, and aspect of the underlying terrain as detailed in Noël et al. (2016)²⁹. For instance, the albedo correction locally enhances surface melt over darker blue ice areas. As blue ice currently remains spatially limited in Antarctica, albedo correction only contributes ~1/15th (3%) to the total surface melt increase obtained after downscaling (+46%). Finally, statistical downscaling estimates elevation gradients locally (using the current and at least five adjacent grid-cells) and at high temporal resolution (daily). This preserves the modelled spatial patterns and synoptic timescales enabling to capture melt and accumulation spatiotemporal variability due to local surface slope, distance to the coast, and atmospheric circulation direction and strength affecting e.g., foehn events. Sources of uncertainty that persist in the current statistical downscaling routine stem from local biases in: a) the modelled elevation gradients (RACMO2.3p2^{17,18} or CESM2²¹) notably in regions characterized by steep slopes and sharp gradients, b) the high-resolution surface topography (REMA)²⁸ used for the elevation correction, c)

the ice sheet (IMBIE-2)², glaciers (RGIv6)⁵⁷ and ice shelf masks (Bedmap2)⁵⁶ used for spatial integration, and d) the satellite albedo product (MODIS) used for ice albedo correction.

Supplementary Figure 7: Spatial correlation with elevation. Spatial correlation between annual modelled SMB and its components averaged for 1979–2021 and surface elevation prescribed in RACMO2.3p2 at 27 km. SMB and components include a) SMB, b) total precipitation, c) surface melt, d) total sublimation, e) drifting snow erosion and f) runoff. Spatial correlation is estimated using annual cumulative values over the current and its eight adjacent grid-cells. A positive/negative correlation means that the SMB component increases/decreases with elevation. Note that in the present-day climate, runoff is spatially limited with high local negative correlations on Larsen B and C, Amery, and Wilkins ice shelves.

2. Lines 74–76: “In particular, accumulation in the Antarctic Peninsula is enhanced on the western windward side of the mountain divide, and subsequently reduced on the eastern lee-side including the Larsen C ice shelf (Fig. 1f), in line with the dry foehn winds that prevail in this region.” I’m not sure that I agree with this description. To me, Fig. 1f shows enhanced accumulation in the 2 km product above the crest of the Peninsula, with reductions relative to the 27 km model at lower levels on both sides of the Peninsula. **We agree with this observation, and reformulated as follows:** “In particular, accumulation is enhanced above the crest of the Antarctic Peninsula, with reductions relative to the 27 km grid at lower levels on both sides of the Peninsula and over Larsen C ice shelf (Fig. 1f), which is in line with the dry foehn winds that prevail in this region³⁶.”

3. Line 79: What are the heavy black lines on supplementary fig. 2a? There is no explanation of them in the figure caption. **This is now clarified in Supplementary Fig. 2a caption:** “In-situ SMB sites (dots) that are situated very close to each other form thick black lines (overlapping dots).”

4. Line 84: “(sub-)Antarctic islands” is a confusing term. “Subantarctic islands” are generally defined as islands north of 60°S but south of the oceanic polar front (e.g. South Georgia). The islands included in your study would generally be classed as Antarctic islands. **Thank you, we replaced “(sub-)Antarctic islands” by “Antarctic islands”.**

5. Lines 152-153: “No significant elevation difference is found over the flat floating ice shelves, except for local crevasses that are resolved on the 2 km grid.” However, there are some quite large differences in melt between the 27 km and 2 km grids across Larsen C, King George VI, Wilkins etc. If elevation here is the same at 27km and 2km, what is driving these differences? Good point, we clarified as follows: “Small negative elevation differences are generally found over the relatively flat floating ice shelves (Supplementary Fig. 1a). Combined with the strong surface melt gradients found nearby the grounding lines, this leads to an overall melt increase that locally peaks over crevasses that are not resolved at 27 km (Figs. 4c). In contrast, the Amery ice shelf shows a small positive elevation difference at 2 km, leading to a local melt decrease compared to the 27 km grid (Figs. 4c and Supplementary Fig. 1a). Locally, the albedo correction implies that surface melt further increases in regions where blue ice is exposed.”

6. Lines 324-325: “To that end, we use a 2 km MODIS 16-day product averaged for the period 2000-2021.” This may be appropriate for the present-day study, but it may become increasingly inaccurate into the future as melt regions expand. Here, we prescribe the present-day MODIS “blue ice” albedo with a base value of 0.55 when firn retreats to expose blue ice both in RACMO2.3p2 at 27 km and at 2 km. This assumption loses validity in case future albedo is e.g., influenced by impurities, which would result in surface darkening (lower albedo), making our approach conservative. This is discussed in the new section *Limitations and uncertainties* as: “In addition, we prescribe the present-day MODIS “blue ice” albedo with a base value of 0.55 when firn retreats to expose blue ice both in RACMO2.3p2 at 27 km and at 2 km. This approach remains valid in the absence of future surface darkening due to biological growth (algae)⁵⁹ or impurity deposition (black carbon, dust, ashes)⁶⁰ as e.g., observed in Greenland. To our knowledge, ice darkening is not yet ongoing in Antarctica. In view of the above, our projected, statistically downscaled surface melt rates should be interpreted as conservative estimates.”

7. Lines 329-331: Isn’t this just repeating what you have already said in line 315, or am I misunderstanding it? You are right, we reformulated as: “Statistical downscaling of total precipitation is motivated by the fact that the native RACMO2.3p2 underestimates orographic precipitation, ...”

Reviewer #2 (Remarks to the Author):

Statistical downscaling at 2km resolution is an effective tool to produce estimates of Antarctic surface mass balance (SMB) and melt in the present-day and future. It improves upon models with coarser spatial resolution by better capturing the topography of the Antarctic ice sheet, particularly on the Antarctic Peninsula. Over the historical period, statistical downscaling at 2km increases SMB by 2% over the Antarctic Ice Sheet compared to 27km resolution RACMO2.3p2. When this downscaled SMB is combined with discharge using the input-output method from 1979-2021, the calculated mass balance is much closer to remote sensing observations of mass balance from GRACE. Also in the present day, it is found that RACMO2.3p2 underestimates surface melting near the grounding line, due to an overestimate of elevation. In low, middle, and high-end projections for the future climate, the systematic underestimate of melt persists in the 27km product compared to the 2km.

Will the work be of significance to the field and related fields? How does it compare to the established literature? The work is original and highlights the important impacts of elevation on modeled SMB and surface melt. Most climate models have a spatial resolution on the order of 10–100km, and this study emphasizes that these grids cannot resolve the complex topography of the Antarctic Ice Sheet, building on Lenaerts et al. (2019). This leads to an underestimation of accumulation on the windward side of mountainous regions and an underestimation of melt near the grounding line. The paper contextualizes the research gap by discussing results from Mottram et al. (2020) and Hansen et al. (2022), which show large variation in annual SMB estimates among multiple regional climate models linked to their grid resolutions. It builds on Dunmire et al. (2022), which evaluates SMB and melt in CESM2 in the historical period and examined future changes under three projected warming scenarios, providing important context for this work. Both Dunmire et al. (2022) and Kittel et al. (2021) highlight the contrast between melt and SMB on the grounded Antarctic Ice Sheet vs ice shelves in future climate scenarios, and this work expands on these studies by closely examining melt patterns along the grounding line at 2km resolution. The results are in agreement with previous studies on regional present and future SMB and melt in Antarctica, for example Donat-Magnin et al. (2021). While prior studies like Gilbert and Kittel (2021) have quantified Antarctic-wide future melt and SMB, this work does so at unprecedented spatial detail, enabling the authors to highlight the topographical effects missed in coarser resolution grids. Thank you!

Does the work support the conclusions and claims, or is additional evidence needed? The paper is clear

regarding its scope and aims, explicitly stating that the primary purpose of statistical downscaling is to correct SMB components for elevation biases between the coarse topography of RACMO2.3p2 (27km) and the 2km REMA topography. Strong evidence is provided to support the claims, including the statistical significance of results comparing the 27km product to the 2km product, uncertainty metrics for SMB and discharge, and thorough comparisons to multiple observational datasets. The results will serve as a step forward in moving the field towards using Antarctic SMB products with finer spatial resolutions to examine the past, present, and future climate. Thank you for the kind words!

Are there any flaws in the data analysis, interpretation, and conclusions? Do these prohibit publication or require revision? The authors present context for uncertainties in modelled SMB among multiple regional climate models in the introduction, and thoroughly compare modelled SMB and melt to observations during the historical period in the results. However, I think this paper would benefit from additional discussion on the uncertainties and biases that can be introduced or enhanced by the statistical downscaling method, in reference to existing biases in RACMO2.3p2, and how much these may influence the result – as well as in the future simulations, which rely on CESM2 and then RACMO2.3p2 prior to the downscaling. The revised manuscript now includes new Supplementary Figure 7 and subsection *Limitations and uncertainties* in the Methods. This new subsection now 1) better supports our choices for elevation as main statistical downscaling predictor, 2) discusses the benefits of estimating elevation gradients at high spatial and temporal resolution in the downscaling approach, and 3) lists remaining sources of uncertainties. See specific comment #1 of Reviewer #1.

Additionally, the methods describe that the land-ice dynamics component of CESM2 is not active, including changes in ice sheet thickness. This implies that SMB and melt-elevation feedbacks due to dynamical changes the ice sheet are not accounted for in the future scenarios. The authors showed evidence of the critical importance of elevation in producing snowfall and melt estimates for Antarctica, especially the Antarctic Peninsula. If the future projections do not account for changing ice sheet elevations in a warming climate, how might that impact the projected melt scenarios presented in the results? RACMO2.3p2 uses a fixed present-day topography and ice geometry. Furthermore, the CESM2 scenario projections used to force RACMO2.3p2 did not have ice dynamics activated (Van Wessem et al., 2023), and to our knowledge, none of the available CMIP6 CESM2 future projections currently have ice dynamics activated for the AIS (Dunmire et al., 2023). In the revised manuscript, we discuss this limitation and potential impacts on our results in the new *Limitations and uncertainties* section as: “RACMO2.3p2^{17,18} and none of the CESM2 scenarios²¹ currently available implement the effect of ice dynamics in Antarctica, i.e., fixed present-day conditions were prescribed in our future projections. Under climate warming, the AIS and floating ice shelves are expected to retreat, calve and/or thicken/thin, which would affect the circulation and associated precipitation distribution as well as melt patterns. For instance, marginal ice thinning could trigger melt-elevation feedback, further enhancing the projected surface melt rates, while snowfall-driven ice thickening could reduce melt. These competing mechanisms will be investigated by statistically downscaling future melt projections from forthcoming fully coupled earth system models that include Antarctic ice dynamics when these become available, as was previously done for the Greenland ice sheet³⁵.”

Regarding the linear regression used to extend the discharge time series to 2021 (P3 L100), is there a precedent for this method? We deem our approach appropriate given the significance of the linear regression (\$R^2=0.95\$, \$p\$ -value \$<0.01\$ ). It is now textually and visually clarified that solid ice discharge data after 2017 were extrapolated (see new Figure 3 below). This is now also clarified in the Methods as: “Although potentially missing discharge events after 2017, using a linear regression is most appropriate provided its high statistical significance (\$R^2=0.95\$, \$p\$ -value \$< 0.01\$ ), and the fact that AIS mass change variability is primarily driven by SMB processes⁶.”

From Figures 2a and 3, it is clear that extending the discharge time series provides more data for comparison to GRACE-observed mass change, and that SMB is the primary driver for short-term variability in the mass balance during the 2018-2021 period, while discharge experiences a more linear increase. However, because the discharge and thus the mass change data are synthetic beyond 2017, I think it would be important to indicate this in the figures. I would strongly recommend adapting the line style, changing the shading, or adding a vertical line in the timeseries to indicate the period when discharge data are extrapolated. Good point, we clarified this by displaying mass change over 2018-2021 as dashed lines. Figures 3 and S4 caption have been modified accordingly: “As the solid ice discharge data set does not extend beyond 2017, it is linearly extrapolated thereafter. The resulting MB at both resolutions is shown as dashed lines after 2017.”

Neither of these points prohibits publication. I would recommend minor revisions to address the uncertainty, the presentation of discharge extrapolated data, and minor comments below. Thank you, we hope that our responses address your remaining comments.

Is the methodology sound? Does the work meet the expected standards in your field? Is there enough detail provided in the methods for the work to be reproduced? The methodology is sound and meets the expected standards in my field. The methods section is detailed, enabling reproducibility of the work regarding the use of RACMO2.3p2, CESM2, and observations from the AntSMB data set, GRACE, and melt records from AWS. Please see limitations below regarding the statistical downscaling method. Thank you!

Limitations My primary limitation as a reviewer is that I do not have expertise in statistical downscaling. I cannot comment on the approach nor the reproducibility of that part of the method. We hope that the revised Methods section is sufficiently detailed, now including a new subsection on uncertainties. For a full description of the downscaling procedure, we refer to Noël et al. (2016).

Minor Comments

P1 L4: “Surface processes are thus essential to quantify AIS mass change” groups ice shelf viability in with mass change; I would recommend either mentioning that ice shelf collapse can accelerate discharge or more distinctly separate ice shelf viability from mass changes. We reformulated as: “..., while meltwater ponding can trigger ice shelf collapse potentially accelerating discharge. Surface processes are thus essential to ...”

P1 L12: It looks like the absolute melt increase is much greater in the high-end warming scenario than the low and middle scenarios (particularly on the Antarctic Peninsula and ice shelves in Supplementary Figure 6), so perhaps it would be better to use “in all scenarios” instead of “irrespective”. We replaced by “in all”.

P2 L41: “general melt underestimation in a regional climate model” – since multiple regional climate models are mentioned earlier in the paragraph, specificity would improve clarity here. Does this refer to RACMO2.3p1 and RACMO2.3p2? We clarified as: “... in e.g., the regional climate model RACMO2.3p2¹⁸.”

P3 L66 and Figure 1f: It would be helpful if the difference in the 2km and 27km ice masks at the margins could be displayed in Figure 1f (as an outline of the 27km ice mask, or hatching, etc.), especially since the effect is highlighted in the text. It is clear to see the resolution difference in Figure 1e, but it is less clear in the difference plot between 2km and 27km. To avoid Figure 1 becoming too crowded, we added two subpanels in Supplementary Figure 1 including the ice mask difference between both resolutions. We refer to Supplementary Figures S1 c,d where appropriate and modified the caption accordingly: “c and d Difference in ice mask between the 2 km and 27 km grids. The common ice mask is shown in grey, added and removed ice pixels in the 2 km grid are displayed in blue and red, respectively. Black contour lines in a-d represent the ice sheet and ice shelves extent at 2 km.”

P3 L70 and Supplementary Figure 1: “2km – 27km” could be added at the top of each plot in Supplementary Figure 1, like in Figures 1c and 1f, to improve clarity. Done.

P3 L83: Including both the definition of ANT and the results describing an increase in SMB from the 27km product to the 2km product forms a very long sentence here – would recommend simplifying or splitting in two. We split the sentence into two as: “We integrate SMB over the whole of Antarctica (ANT), including the grounded AIS, adjacent Antarctic islands, and floating ice shelves. Accumulation in the 2 km product ...”.

P3 L85 and Figure 2: I find the colors used in Figures 2a and 2c slightly confusing, because almost the same 3 colors are used in each figure to represent different things. For example, to me it is non-intuitive that SMB AIS is portrayed in red in 2a and Melt AIS is portrayed in red in 2c, while at the same time SMB ANT is in black in 2a and Melt ANT is in yellow in 2c. It would be helpful if the authors could explain the reasoning for these color choices in a response or modify the colors to improve the consistency of color choices between the two figures. To clarify, we show AIS data in red (a SMB, b melt), ANT data in black, and observational data in orange (a Discharge, b QuickSCAT). For consistency, discharge data extrapolated beyond 2017 are shown as a dashed line. We modified the figure caption accordingly.

P5 L143 and Figure 4f: As mentioned previously regarding Figure 1f, it would also be helpful if the difference in the 2km and 27km ice masks at the margins could be displayed in Figure 4f. The ice mask difference is now shown in Supplementary Fig. 1c, d.

P5 L159 and Supplementary Figure 2a: I count 10 yellow stars in the figure (6 in DML), while 9 are mentioned in the text. Also, the stars are difficult to spot in the figure, can you increase their size and/or change the color to make them stand out better? Thank you for spotting this, indeed there were 10 instead of 9 sites. This is corrected, and we increased the size of the yellow stars.

P5 L178: “increase in melt over floating ice shelves”. Done.

P12 L401: “(orange band in Figure 2d)”, “(black line in Figure 2d)”, and “(grey band in Figure 2d)” – should this be referring to 2c? This is corrected.

References Cited in the paper and in the reviewer comments:

- Dunmire et al. (2022), Hansen et al. (2022), Lenaerts et al. (2019), Mottram et al. (2020).

Additional references:

- Donat-Magnin, Jourdain, Kittel, Agosta, Amory, Gallée, Krinner, and Chekki (2021). “Future surface mass balance and surface melt in the Amundsen sector of the West Antarctic Ice Sheet”, *The Cryosphere*, doi: 10.5194/tc-15-571-2021
- Gilbert and Kittel (2021). “Surface Melt and Runoff on Antarctic Ice Shelves at 1.5deg C, 2deg C, and 4deg C of Future Warming”, *Geophysical Research Letters*, doi: 10.1029/2020GL091733
- Kittel, Amory, Agosta, Jourdain, Hofer, Delhasse, Doutreloup, Huot, Lang, Fichet, and Fettweis (2021). “Diverging future surface mass balance between the Antarctic ice shelves and grounded ice sheet”, *The Cryosphere*, doi: 10.5194/tc-15-1215-2021

Reviewer #3 (Remarks to the Author):

This is a very clear and well written manuscript describing a new downscaling of SMB in Antarctica using higher resolution than existing current state of the art downscaled products. The implications of higher resolution downscaling are well described and are important. The authors describe in particular differences that appear in the total surface mass balance integrated over Antarctica, and also differences locally, in particular in the Antarctic Peninsula where topographic gradients are most important. In addition, the authors demonstrate clearly how the new downscaling is able to explain offsets in previous downscaling between remote sensing data and reconstruction mass balance time series. This was particularly important, as a simple comparative study between low and higher-order resolution would have been difficult to validate. The implications were also clearly explained in terms of projections and how this could impact the way surface melt-rates should be computed for increasingly warm scenarios at 2100 and beyond. I was very intrigued by the impact at the grounding line in particular, though I thought the authors put too strong an emphasis on hydrofracturing when describing the implications for ice shelf stability. More could be said about the impact on runoff in particular which has been found to be significantly disruptive to regional ocean circulation, a controlling factor of basal melt rates, another key component of the stability of ice shelves. I suggested changes to the wording in more details (see attached commented pdf). A little bit more effort should also be carried out in terms of anchoring this study against other existing downscaling efforts, maybe for example drawing from studies done at lower resolution for Greenland? Not being an SMB expert, I would like to understand how this anchored into the existing literature. Overall, I think this is a worthy scientific addition to our knowledge, and should be shared with

the Cryospheric community, I therefore recommend it be published following minor revisions to address in particular the way the study is introduced and anchored in the existing literature. Thank you, we grouped your three main comments and our replies below:

1. More could be said about the impact on runoff in particular which has been found to be significantly disruptive to regional ocean circulation, a controlling factor of basal melt rates, another key component of the stability of ice shelves. Thank you for providing this insight, in the revised manuscript we now mention runoff as potentially important for ocean circulation and ice shelf stability: “As for Greenland fjords, (future) increase in surface and subsurface runoff have the potential to enhance sub-shelf basal melting and subsequent destabilisation by thinning ice shelves from below⁴. These combined processes, i.e., surface meltwater ponding and basal melt, can trigger ice shelf disintegration by hydrofracturing, a mechanism that reduces the buttressing effect on the grounded AIS to eventually accelerate solid ice discharge and sea-level rise¹³.”
2. I suggested changes to the wording in more details (see attached commented pdf). Please, find our responses below.
3. A little bit more effort should also be carried out in terms of anchoring this study against other existing downscaling efforts, maybe for example drawing from studies done at lower resolution for Greenland? Not being an SMB expert, I would like to understand how this anchored into the existing literature. In the revised manuscript, we now introduce past SMB downscaling studies: “The ability of statistical downscaling to refine the spatial distribution of SMB components was first demonstrated for the Greenland ice sheet, where the downscaled product realistically captured high mass loss rates over narrow ablation zones and outlet glaciers that are typically unresolved in RACMO2.3p2²⁹. Likewise, statistical downscaling to (sub-)kilometer spatial resolution proved essential to accurately quantify contemporary (and projected) mass change of the Greenland ice sheet^{30,31}, its peripheral ice caps³², glaciers of the Canadian Arctic³³, Svalbard³⁴, and Iceland³⁵, and their contribution to global sea-level rise.”

Point comments:

L4 Among other factors. Melt ponding indeed, but in particular basal melt rates, and rifting/cracking. See our response to comment #1.

L24 Again I agree, but let's not forget the bottom of the ice shelves, plenty of processes here that could do the same as hydrofracturing. See our response to comment #1.

L25 Debatable on Larsen A, which was heavily fractured prior to ponding. Larsen B indeed was ponding prior to disintegration. Thank you, we removed the reference to Larsen A: “This has previously occurred in the Antarctic Peninsula over Larsen B ice shelf (March 2002)¹⁴.”

L41 Nice transition. Thank you, note the suggestion from Reviewer #2: “... in e.g., the regional climate model RACMO2.3p2¹⁸.”

L48 Do not assume that the community understand regional downscaling. Please explain it with 1-2 sentences of introduction. Especially that you have RACMO output, ERA5 reanalysis in the past, CESM2 in the future, who is forcing what? who is regional, who is global? Thank you, we reformulated as: “Here we present new, daily SMB and surface melt products covering the grounded AIS and floating ice shelves at 2 km spatial resolution for the contemporary climate and three scenario projections until 2100. As a first step, present-day climate from the global climate reanalysis ERA5²⁶ (1979-2021) and three global climate projections from the Community Earth System Model (CESM2)²⁷ under a low (SSP1-2.6), moderate (SSP2-4.5), and high-end (SSP5-8.5) warming scenario (1950-2099) are used as lateral forcing for the Regional Atmospheric Climate Model (RACMO2.3p2)^{17,18}, which simulates the contemporary and future SMB (components) of Antarctica on a 27 km grid (see Methods). As a second step, statistical downscaling is applied to correct these SMB components ...”

L52 I would put this statement earlier on in the introduction, would make the article more readable earlier on. We modified the text as suggested in the comment in L48. We hope that our changes also address this comment.

L65 Fig1 You could reduce the number of insets here, because i honestly can't tell the difference between Antarctica at 2 or 27km, but I can definitely see on the diff image what is going on. It would declutter your figures, which have a lot of content. Although we see your point, we also feel that it is important to show both the low-resolution model and the high-resolution downscaled products to visualize the spatial improvements. This is less evident on a difference plot, so we kept Figures 1 and 4 as are.

L101 Do you really need to do this? To quantify mass change estimates beyond 2017, we needed a relevant way to extend the solid ice discharge times series. A linear regression was the most relevant approach given its high statistical significance ($R^2=0.95$, $p\text{-value}<0.01$). This is now also clarified in the Methods as: “Although potentially missing discharge events after 2017, using a linear regression is most appropriate provided its high statistical significance ($R^2=0.95$, $p\text{-value} < 0.01$), and the fact that AIS mass change variability is primarily driven by SMB processes⁶.” We followed Reviewer #2 suggestion to display the extrapolated D data as dashed lines.

I understand scientists want the latest and greatest, but I can accommodate the estimates until 2017 instead of 2021. Why not then until 2023? Our GRACE mass change estimates do not currently extend to 2023, the (downscaled) RACMO2.3p2-ERA5 product is not extended to 2023 yet.

I think taking the risk of a linear regression won't add to your paper here. You could miss big events in D by doing this (ex: calving events, increases in calving, as demonstrated recently by Greene et al, 2023). We agree that the approach could miss discharge events but deem that our approach, in the absence of updated and published discharge data, still provides relevant estimates. See our answer to comment in L101.

L136 if you talk about ice shelf melt rates, to me this means basal. I would pay attention to avoiding biasing the writeup of the paper too much to the surface community. This is a nice paper, very clear, but there are trigger words here that make me think a lot about the bottom of the ice shelves, when it's not the goal of the paper. Thank you, we clarified this where appropriate. The term “surface melt” now appears in the revised title: “Higher Antarctic ice sheet accumulation and surface melt rates revealed at 2 km resolution”

L137 I can think also of water runoff completely modifying ocean circulation under ice shelves, consequentially impacting basal melt rates, and sometimes restabilizing an ice shelf (Poinelli et al, 2022). Or interactions of rifts with ocean circulation. So be careful in accepting hydrofracturing as the only process destabilizing/stabilizing ice shelves. To clarify, we start the sentence with: “Among other processes⁴, ...”. While keeping the study focused and concise, this avoids digressions from our main message on surface melt and SMB.

L144 This is one of the points you are raising that worry me. There are significant changes in calving front position in the Peninsula. At 2 km, they are well captured, at 27 km, not so much. Unless I missed it, I would like to a discussion of how the 2 km and 27 km masks were built. Good point. RACMO2.3p2 ice mask and topography at 27 km are down-sampled from the 1 km resolution digital elevation model of Bamber et al. (2009). We clarified

as: “Surface topography is prescribed from the 1 km digital elevation model of Bamber et al. (2009)⁴⁷, down-sampled to 27 km.” The Methods section provides resources used in the downscaled product, which combines the ice sheet mask from IMBIE-2, land ice on Antarctic islands from RGIv6, mask of floating ice shelves from Bedmap2, and the digital elevation model REMA down-sampled to 2 km.

Were they changing in time, which would impact your conclusions I would guess. We distinguish two ice mask changes: (1) those that result from the downscaling procedure (27 km to 2 km), (2) those resulting from a changing Antarctic ice sheet mask in a future warmer climate. The ice mask extent difference referred to in the text is the former (1), which is now made explicit in new Supplementary Figures 1c,d (see below). Regarding the climate-related change, note that neither RACMO2.3p2 nor any of the available CESM2 scenarios had Antarctic ice dynamics activated, i.e., ice mask and topography were kept fixed in the present-day and future projections. To clarify, we added the following to the text of the Methods section: “In the absence of interactive ice dynamics in RACMO2.3p2, ice mask and topography are kept fixed.”

L175 Again I would like to know whether the mask is dynamic. If fixed, how does this work? How much mass are you losing at the edges. For ISMIP6, this was a significant concern when we used RACMO for mass balance assessments. We have to figure out a strategy to avoid losing mass between intercompared model results. I need a little bit more explanation and numbers on the transient mask evolution and the impact on your comparisons. See our previous response, the ice mask is not dynamic, i.e., ice is fixed and cannot retreat, thicken/thin, or calve. The main reason for doing this is that ice dynamics is not included in any of the models used in this paper (and in CMIP6 for that matter). The only ice mask difference discussed in the paper is that occurring when downscaling the 27 km grid onto the 2 km grid, see new Supplementary Figures 1c,d (also presented above). In the revised manuscript, we now elaborate on this in the new *Limitations and uncertainty* section: “RACMO2.3p2^{17,18} and none of the CESM2 scenarios²¹ currently available implement the effect of ice dynamics in Antarctica, i.e., fixed present-day conditions were prescribed in our future projections. Under climate warming, the AIS and floating ice shelves are expected to retreat, calve and/or thicken/thin, which would affect the circulation and associated precipitation distribution as well as melt patterns. For instance, marginal ice thinning could trigger melt-elevation feedback, further enhancing the projected surface melt rates, while snowfall-driven ice thickening could reduce melt. These competing mechanisms will be investigated by statistically downscaling future melt projections from forthcoming fully coupled earth system models that include Antarctic ice dynamics when these become available, as was previously done for the Greenland ice sheet³⁵.”

L182 I'm struggling to understand why? Is it katabatic winds that are better modeled? Pure downscaling effects? It would be nice to understand why, because the implications for grounded line dynamical retreat are significant potentially. We broke down the components of the surface melt increase in the revised manuscript, where it is now described as follows: “The elevation correction contributes 38 Gt yr⁻¹ (34%) to the total surface melt increase (Supplementary Fig. 1a,b). This effect is particularly important near the grounding line where surface

elevation is generally reduced at 2 km and steep topographic gradients were not accurately captured at 27 km. Over low-lying ice shelves, the combined elevation difference and strong melt gradients locally enhance surface melt at 2 km. Spatial refinement of the ice mask from 27 km to 2 km contributes 10 Gt yr^{-1} (9%) to the total melt increase (Supplementary Fig. 1c,d), while the remaining 3 Gt yr^{-1} (3%) stem from albedo correction over blue ice areas.”

L193 sorry to harp on this, you need to consistently change this to "surface melt", or clearly explain somewhere in the introduction that you are never referring to basal melt rates. Being a dynamicist, to me $dh/dt = \text{div } H_u + \text{SMB} + \text{BMB}$, so ice thinning is controlled by surface and basal melt rate as well as flux divergence. You are "only" dealing with 1 or 4 processes here. Avoiding confusion is necessary I think to clarify the impact of your paper. Thank you, we adjusted the manuscript accordingly.

L211 Very nice. Thank you!

Figures I can barely read this, even on my large screen with magnification. This is not readable in A4 format. Please increase font size for every image. We increased the font size when possible.

REVIEWERS' COMMENTS

Reviewer #1 (Remarks to the Author):

The authors have responded positively to comments that I made on an earlier version of this manuscript and I am now happy to recommend publication of the paper without further revision.

Reviewer #2 (Remarks to the Author):

The authors have addressed my comments in a satisfactory manner. In particular, the authors added a Limitations and Uncertainties section in the methods, which addresses my primary comments about uncertainties and contextualizes their results in the current state of Antarctic climate modeling. I am happy to recommend publication of the manuscript in its current form.

Response letter: Higher Antarctic ice sheet accumulation and surface melt rates revealed at 2 km resolution.

Reviewer #1 (Remarks to the Author):

The authors have responded positively to comments that I made on an earlier version of this manuscript and I am now happy to recommend publication of the paper without further revision.

Thank you very much for your positive comments that helped us improve the quality and robustness of our manuscript.

Reviewer #2 (Remarks to the Author):

The authors have addressed my comments in a satisfactory manner. In particular, the authors added a Limitations and Uncertainties section in the methods, which addresses my primary comments about uncertainties and contextualizes their results in the current state of Antarctic climate modeling. I am happy to recommend publication of the manuscript in its current form.

Thank you very much for your positive comments that helped us improve the quality and robustness of our manuscript.